# Structured Multi-Track Accompaniment Arrangement via Style Prior Modelling

**Jingwei Zhao**[1,3]     **Gus Xia**[4,5]     **Ziyu Wang**[5,4]     **Ye Wang**[2,1,3]

[1]Institute of Data Science, NUS     [2]School of Computing, NUS
[3]Integrative Sciences and Engineering Programme, NUS Graduate School
[4]Machine Learning Department, MBZUAI     [5]Computer Science Department, NYU Shanghai
jzhao@u.nus.edu   gus.xia@mbzuai.ac.ae
ziyu.wang@nyu.edu   wangye@comp.nus.edu.sg

## Abstract

In the realm of music AI, arranging rich and structured multi-track accompaniments from a simple lead sheet presents significant challenges. Such challenges include maintaining track cohesion, ensuring long-term coherence, and optimizing computational efficiency. In this paper, we introduce a novel system that leverages *prior modelling over disentangled style factors* to address these challenges. Our method presents a two-stage process: initially, a piano arrangement is derived from the lead sheet by retrieving *piano texture* styles; subsequently, a multi-track orchestration is generated by infusing *orchestral function* styles into the piano arrangement. Our key design is the use of vector quantization and a unique multi-stream Transformer to model the *long-term* flow of the orchestration style, which enables flexible, controllable, and structured music generation. Experiments show that by factorizing the arrangement task into interpretable sub-stages, our approach enhances generative capacity while improving efficiency. Additionally, our system supports a variety of music genres and provides style control at different composition hierarchies. We further show that our system achieves superior coherence, structure, and overall arrangement quality compared to existing baselines.

## 1   Introduction

Representation learning techniques have enabled new possibilities for controllable generative modelling. By learning *implicit* style representations, which are often hard to explicitly label (*e.g.*, timbre of music audio [21], texture of music composition [39], and artistic style in paintings [20]), new music and artworks can be created via style transfer and latent space sampling. These learned style factors can also serve as external controls for downstream generative models, including Transformers [18, 36] and diffusion models [42]. However, applying style factors to *long-term* sequence generation remains a challenging task. Existing approaches rely on style templates specified manually or by heuristic rules [36, 42, 51], which are impractical for long-term generation. Moreover, when structural constraints are imposed, misaligned style factors can result in incoherent outputs.

To address these challenges, we aim to develop a novel sequence generation framework leveraging a global style planner, or *prior*, which models the conditional distribution of *style factors* given the model input's *content factors*. Both style and content factors are sequences of compact, structurally aligned latent codes over a disentangled representation space. By infusing the style back to the content, we can recover the observational target with globally coherent style patterns.

In this paper, we study *style prior modelling* through the task of *multi-track accompaniment arrangement*, a typical scenario for long-term conditional sequence generation. We assume the input of a

---

piano accompaniment score, which typically carries a verse-chorus structure. Our target is to generate corresponding multi-track arrangements featuring band orchestration. We start by disentangling a band score at time $t$ into *piano reduction* $\mathbf{c}_t$ (content factor) and *orchestral function* $\mathbf{s}_t^k$ (style factors for individual tracks $k = 1, 2, \cdots, K$). On top of this, we model the prior of *finding appropriate functions to orchestrate a given piano score*, or formally $p(\mathbf{s}_{1:T}^{1:K} \mid \mathbf{c}_{1:T})$. To model dependencies in both time ($T$) and track ($K$) directions, we develop a multi-stream Transformer with interleaved time-wise and track-wise layers. The track-wise layer allows for flexible control over the choice of instruments and the number of tracks, while the time-wise layer ensures structural alignment through cross-attention to the *piano reduction*. Decoding the inferred $\mathbf{s}_{1:T}^{1:K}$ with $\mathbf{c}_{1:T}$, we can address accompaniment arrangement in a flexible *multi-track* form with extended *whole-song* structure.

Experiments show that our method outperforms existing sequential token prediction approaches and provides better multi-track cohesion, structural coherence, and computational efficiency. Additionally, compared to existing designs of multi-stream language models, our model handles flexible stream combinations more effectively with enhanced generative capacity.

To summarize, our contributions in this paper are three-folded:

- We propose **style prior modelling, a hierarchical generative methodology** addressing both long-term structure (via style prior at high level) and fine-grained condition/control (via representation disentanglement at low level). Our approach moves beyond the limitation of manual specification of style factors, providing a flexible, efficient, and self-supervised solution for long-term sequence prediction and generation tasks.

- We propose a novel **layer interleaving architecture** for multi-stream language modelling. In our case, it models parallel music tracks with a flexible track number, controllable instruments, and manageable computation. To our knowledge, it is the first multi-stream language model with tractable generalization to flexible stream combinations.

- Integrating our previous study on *piano texture* style transfer [39, 50], we present a **complete music automation system** arranging an input lead sheet (a basic music form with melody and chord only) via piano accompaniment to multi-track arrangement. The entire system is interpretable at two composition hierarchies: 1) *piano texture* and 2) *orchestral function*, and demonstrates state-of-the-art arrangement performance for varied genres of music.[1]

## 2 Related Works

In this section, we overview three topics related to our study. Section 2.1 reviews existing studies on representation disentanglement. Section 2.2 summarizes prior modelling methods in music generation. Section 2.3 reviews the current progress with the task of accompaniment arrangement.

### 2.1 Content-Style Disentanglement via Representation Learning

Representation disentanglement is a popular technique in deep generative modelling [3, 16, 48, 49]. In the music domain, this approach has proven valuable by learning compositional factors related to music *style* and *content*. By manipulating these factors through interpolation [32], swapping [39], and prior sampling [46], it provides a self-supervised and controllable pathway for various music automation tasks. Recent works leverage disentangled style factors as control signals for long-term music generation [36, 42]. However, these approaches typically treat style representations as fixed condition sequences during training, requiring manual specification or additional algorithms for control during inference. In contrast, we model the prior of *the style to apply* conditional on the given music content, which is a more generalized and flexible approach.

### 2.2 Music Generation with Latent Prior

In sequence generation tasks (*e.g.*, music and audio), learning a prior sampler over a compact, latent representation space is often more efficient and effective. Jukebox [7] models the latent codes encoded by VQ-VAEs [34] as music priors, which can further reconstruct minutes of music audio. More recently, MusicLM [2] and MusicGen [4] learn multi-modal priors for generating music from text

---

[1]Demo and more resources: `https://zhaojw1998.github.io/structured-arrangement/`

Table 1: Summary of the data representations applied in this paper. We use notation $[a..b]$ to denote the integer interval $\{x \mid a \leq x \leq b, x \in \mathbb{Z}\}$ including both endpoints.

| | Multi-Track Arrangement | Piano Reduction | Orchestral Function |
|---|---|---|---|
| **Data Representation** | $\mathbf{x} \in [0..32]^{T \times K \times 32 \times 128}$ | $\mathrm{pn}[\mathbf{x}] \in [0..32]^{T \times 32 \times 128}$ | $\mathrm{fn}[\mathbf{x}] \in [0,1]^{T \times K \times 32}$ |
| **Latent Dimension** | $\mathbf{z} \in \mathbb{R}^{T \times K \times 256}$ | $\mathbf{c} \in \mathbb{R}^{T \times 256}$ | $s \in [0..127]^{8T \times K}$ |

prompts. While prior modelling facilitates long-term generation, the latent codes in these works are not interpretable, thus lacking a precise control by music content-based signals (*e.g.*, music structure). Such controls are essential for conditional generation tasks, including accompaniment arrangement. In this paper, we model a *style prior* conditional on the disentangled music content, which allows for structured long-term music generation, enhancing both interpretability and controllability.

## 2.3   Accompaniment Arrangement

Accompaniment arrangement aims to compose the accompaniment part given a lead sheet, which is a difficult conditional generation task involving structural constraints. Existing methods mainly train a conditional language model based on sequential note-level tokenization [14, 15, 30, 33], which often suffer from slow inference speed, truncated structural context, and/or simplified instrumentation. Recent attempts with diffusion models show higher sample quality with faster inference [23, 26, 27], but still consider limited instruments or tracks. AccoMontage [47, 50] maintains a whole-song structure by manipulating high-level composition factors, but is limited to piano arrangement alone. Our paper presents a two-stage approach: from lead sheet to piano accompaniment, and from piano to multi-track, both leveraging prior modelling of high-level style factors. This approach offers modularity [11] and enables high-quality *whole-song* and *multi-track* accompaniment arrangement.

## 3   Method

We develop a model that takes a *piano reduction* as input and outputs an orchestrated multi-track arrangement. Using an autoencoder, we disentangle a multi-track music score into its *piano reduction* (content factor) and *orchestral function* (style factor). We then design a prior model to infer *orchestral functions* given the *piano reduction*. The autoencoder operates at segment level, while the prior model works on the whole song. The entire model can operate as an orchestrator module in a complete arrangement system. In this section, we introduce our data representation in Section 3.1, autoencoder framework in Section 3.2, and prior model design in Section 3.3.

### 3.1   Data Representation

We summarize our data representations in Table 1. Let $\mathbf{x}$ be a $K$-track arrangement score. We split it into $T$ segments and represent $\mathbf{x}_t^k$ — each segment track — as a matrix of shape $P \times N$. Here $P = 128$ represents 128 MIDI pitches and $N$ is the time dimension of a segment. This matrix representation aligns with the modified piano roll in [39], where each non-zero entry $(p, n) > 0$ indicates a note onset and its value indicates the note duration. In this paper, we primarily focus on music pieces in 4/4 time signature with 1/4-beat resolution. Duration values range from 1 (for sixteenth notes) to 32 (for double whole notes). We consider 1 segment = 8 beats (2 bars) and derive $N = 32$, which is a proper scale for learning music content/style representations [37, 39–41, 46].

The *piano reduction* of $\mathbf{x}$ is notated as $\mathrm{pn}[\mathbf{x}]$. It is approximated by downmixing all $K$ tracks into a single-track mixture similar to [8]. When concurring notes are found across tracks, we keep the one with the largest duration (*i.e.*, track-wise maximum). Segment-wise, $\mathrm{pn}[\mathbf{x}]_t$ is also a $P \times N$ matrix. It preserves the overall music content while discarding the multi-track form.

The *orchestral function* of $\mathbf{x}$ is notated as $\mathrm{fn}[\mathbf{x}]$. It describes the rhythm and grooving patterns [45] of each segment track, which serves as the "skeleton" of a multi-track form. Formally,

$$\mathrm{fn}[\mathbf{x}]_t^k = \mathrm{colsum}(\mathbf{1}_{\{\mathbf{x}_t^k > 0\}})/\mathrm{max\_sum}, \tag{1}$$

where indicator function $\mathbf{1}_{\{\cdot\}}$ counts each note onset position as 1; $\mathrm{colsum}(\cdot)$ sums up the pitch dimension, deriving a $1 \times N$ time-series feature; $\mathrm{max\_sum} = 14$ is for normalization. The *orchestral*

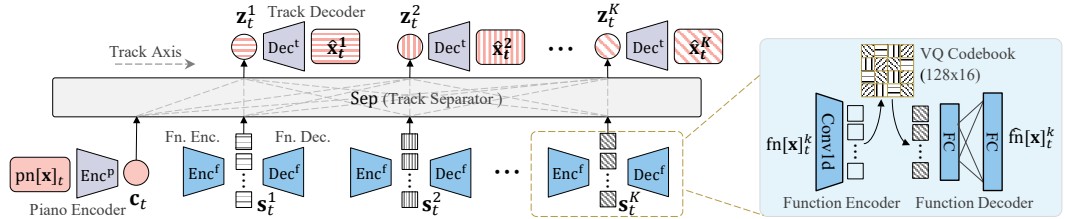

Figure 1: The autoencoder architecture. It learns content representation $c_t$ from *piano reduction*, style representations $\mathbf{s}_t^{1:K}$ from *orchestral function*, and leverages both to reconstruct individual tracks.

*function* $\text{fn}[\mathbf{x}]$ essentially describes the form, or layout, of multi-track music $\mathbf{x}$. It indicates the rhythmic intensity of parallel tracks and informs where to put more notes and where to keep silent.

### 3.2 Autoencoder

Our autoencoder consists of two components as shown in Figure 1. A VQ-VAE submodule (right of Figure 1) encodes *orchestral function* $\text{fn}[\mathbf{x}]_t^k$. A VAE module (left of Figure 1) encodes *piano reduction* $\text{pn}[\mathbf{x}]_t$ and reconstructs individual tracks $\mathbf{x}_t^{1:K}$ leveraging the cues from $\text{fn}[\mathbf{x}]_t^{1:K}$. During training, both inputs $\text{pn}[\mathbf{x}]$ and $\text{fn}[\mathbf{x}]$ are deterministic transforms from the output $\mathbf{x}$ and the entire model is self-supervised. We see similar techniques for representation disentanglement in [39, 41, 46].

The VQ-VAE consists of Function Encoder $\text{Enc}^{\text{f}}$ and Decoder $\text{Dec}^{\text{f}}$. Encoder $\text{Enc}^{\text{f}}$ contains a 1-D convolutional layer followed by a vector quantization block. Our intuition for applying a VQ-VAE is that *orchestral function* commonly consists of rhythm patterns (such as syncopation, arpeggio, *etc.*) that can naturally be categorized as *discrete* variables. In our case, each segment track is encoded into 8 discrete embeddings on a 1-beat scale, indicating the flow of orchestration style. Formally,

$$\mathbf{s}_t^k := \{s_\tau^k\}_{\tau=8t-7}^{8t} = \text{Enc}^{\text{f}}(\text{fn}[\mathbf{x}]_t^k), \ k = 1, 2, \cdots, K, \tag{2}$$

where $s_\tau^k$ is the latent *orchestral function* code for the $k$-th track at the $\tau$-th beat. We encode $\text{fn}[\mathbf{x}]_t^k$ at a finer 1-beat scale (instead of segment) to preserve fine-grained rhythmic details. The new scale is re-indexed by $\tau = 1, 2, \cdots, 8T$. We collectively denote each 8-code grouping as $\mathbf{s}_t^k$ for conciseness.

The VAE consists of Piano Encoder $\text{Enc}^{\text{p}}$, Track Separator $\text{Sep}$, and Track Decoder $\text{Dec}^{\text{t}}$. Encoder $\text{Enc}^{\text{p}}$ learns content representation $c_t$ from *piano reduction* $\text{pn}[\mathbf{x}]_t$. Here $c_t$ is a *continuous* representation (without vector quantization) that captures more nuanced music content. Decoder $\text{Dec}^{\text{t}}$ reconstructs individual tracks $\mathbf{x}_t^k$ from track representation $\mathbf{z}_t^k$. Notably, $\mathbf{z}_t^{1:K}$ are recovered from $c_t$ using the *orchestral function* cues from $\mathbf{s}_t^{1:K}$. Formally,

$$\mathbf{z}_t^1, \mathbf{z}_t^2, \cdots, \mathbf{z}_t^K = \text{Sep}(\mathbf{s}_t^1, \mathbf{s}_t^2, \cdots, \mathbf{s}_t^K \mid c_t), \tag{3}$$

where Track Separator $\text{Sep}$ is a Transformer encoder. In this process, each $\mathbf{s}_t^k$ queries $c_t$ to recover the corresponding track ($k$), while they also attend to each other to maintain the dependency among parallel tracks. Learnable instrument embeddings [51] are added to each track based on its instrument class. We provide details of the autoencoder architecture in Appendix A.1.

### 3.3 Style Prior Modelling

The VQ-VAE in Section 3.2 derives latent codes $s_{1:8T}^{1:K}$ for *orchestral function* as a multi-stream time series. Here $k = 1, 2, \cdots K$ is the stream (track) index and $\tau = 1, 2, \cdots, 8T$ is the time (beat) index. The purpose of *style prior modelling* is to infer *orchestral function* given *piano reduction* so that the former can be leveraged to orchestrate the latter into multi-track music. We design our prior model as shown in Figure 2. It is an encoder-decoder framework that models parallel tracks/streams of *orchestral function* codes conditional on the *piano reduction*.

The decoder module (right of Figure 2) has alternate layers of Track Encoder and Auto-Regressive Decoder. Track Encoder is a standard Transformer encoder layer [35] and it aggregates inter-track information along the track axis. Auto-Regressive Decoder is a Transformer decoder layer (with self-attention and cross-attention) and it predicts next-step *orchestral function* codes on the time axis.

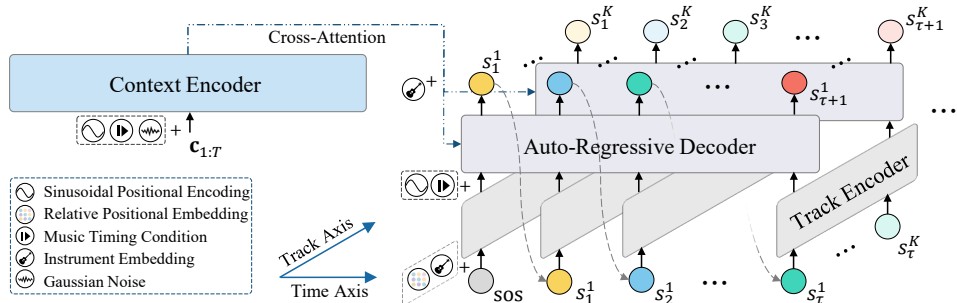

Figure 2: The prior model architecture. The overall architecture is an encoder-decoder Transformer, while the decoder module is interleaved with orthogonal time-wise and track-wise layers.

By orthogonally stacking two types of layers, we can model track-wise and time-wise dependencies simultaneously with a manageable computational cost. Compared to sequential tokenization methods in previous studies [9, 30, 36], our method brings down the complexity from $\mathcal{O}(N^2T^2)$ to $\mathcal{O}(\max(N,T)^2)$. Moreover, we support a flexible multi-track form ($N$ being variable) with a diverse instrumentation option. We add instrument embedding [51] and relative positional embedding [15] to the track axis, where 34 instrument classes [25] are supported. We add music timing condition [7] to the time axis, which encodes the positions in a training excerpt as fractions of the complete song, helping the model capture the overall structure of a song.

The encoder module (left of Figure 2) of our prior model is a standard Transformer encoder, which takes *piano reduction* $\mathbf{c}_{1:T}$ as global context. It is connected to the decoder module via cross-attention and maintains the global phrase structure. During training, both $\mathbf{c}_{1:T}$ and $s_{1:8T}^{1:K}$ are derived from the same multi-track piece and the entire model is self-supervised. Let $p_\theta$ be the distribution of *orchestral function* codes fitted by our prior model $\theta$, the training objective is the mean of negative log-likelihood of next-step code prediction:

$$\mathcal{L}(\theta) = -\frac{1}{K}\sum_{k=1}^{K}\log p_\theta(s_\tau^k \mid s_{<\tau}^{1:K}, \mathbf{c}_{1:T}). \tag{4}$$

We provide more implementation details of the prior model in Appendix A.2. We note that there is a potential domain shift from our approximated *piano reduction* to real piano arrangements. To prevent overfitting, we use a Gaussian noise $\epsilon$ to blur $\mathbf{c}_{1:T}$ while *preserving its high-level structure*. During training, $\epsilon$ is combined with $\mathbf{c}_{1:T}$ using a weighted summation with noise weight $\gamma$ ranging from 0 to 1. It encourages a partial unconditional generation capability. At inference time, $\gamma$ is a parameter that can balance creativity with faithfulness. An experiment on $\gamma$ is covered in Appendix C.

## 4 Whole-Song Multi-Track Accompaniment Arrangement

We finalize a complete music automation system by applying *style prior modelling* at two cascaded stages. As shown in Figure 3, our autoencoder and *orchestral function prior* operate on Stage 2 for *piano to multi-track* arrangement. On Stage 1, we adopt our previous study, a *piano texture prior* [50]

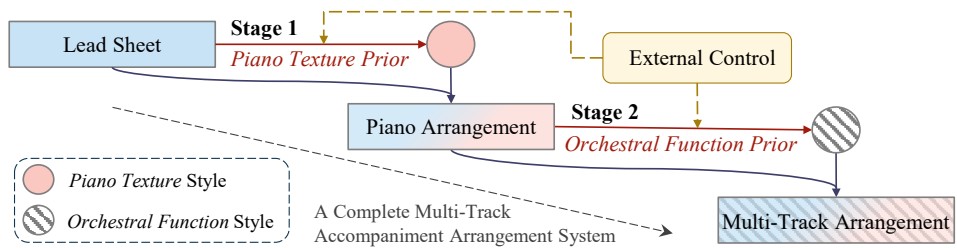

Figure 3: A complete accompaniment arrangement system based on cascaded prior modelling. The first stage models *piano texture* style given lead sheet while the second stage models *orchestral function* style given piano. Besides modularity, the system offers control on both composition levels.

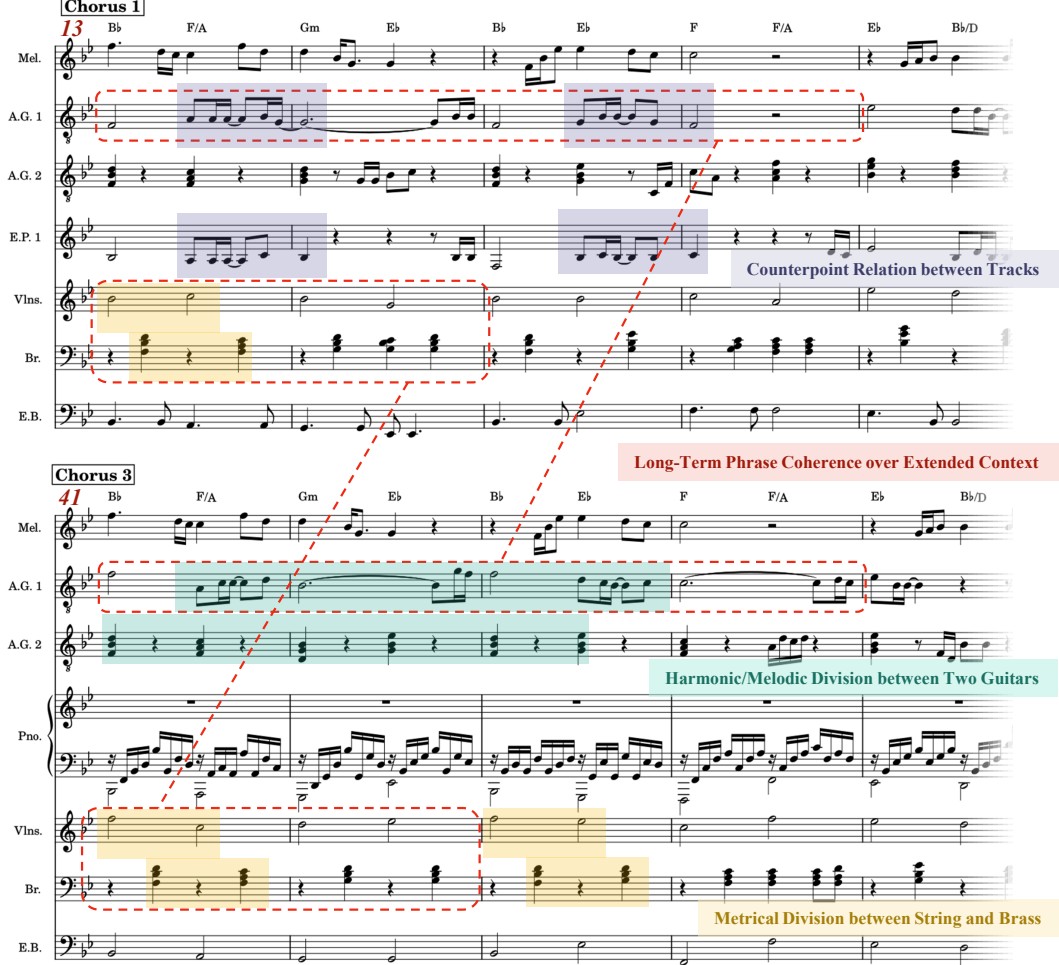

Figure 4: Arrangement for *Can You Feel the Love Tonight*, a pop song in a total of 60 bars. We show two chorus parts from bar 13 to 41. We use red dotted boxes to show coherence in long-term structure. We use coloured blocks to show naturalness and cohesion in multi-track arrangement.

on top of chord/texture representation learning [39], for *lead sheet to piano* arrangement. Given a lead sheet, the first stage generates a piano accompaniment, establishing the rough whole-song structure. Our system then orchestrates the piano accompaniment into a complete multi-track arrangement with band instrumentation. This two-stage approach mirrors musicians' creative workflow [1] and allows for control at both composition levels. In particular, we provide three control options:

1. Texture Selection: To filter *piano textures* on Stage 1 by metadata and statistical features.
2. Instrumentation: To customize the *track number* and *choice of instruments* on Stage 2.
3. Orchestral Prompt: To prompt the orchestration process with an *orchestral function* template.

We showcase an arrangement example by the complete system in Figure 4. The system input is a lead sheet shown by the `Mel` staff. The final output is the accompaniment in the rest staves. Notably, the lead sheet consists of 60 bars in a structure of `i4A8B8B8x4A8B8B8O4` (using notations by [5]). Here, `i4`, `x4`, and `O4` each denote a 4-bar intro, interlude, and outro. `A8` and `B8` represent an 8-bar verse and chorus, respectively. Figure 4 shows the arrangement result for the first and third choruses, spanning from bar 13 to 41. We leverage control option 2 to customize the instrumentation as celesta, acoustic guitars (2), electric pianos (2), acoustic piano, violin, brass, and electric bass in a total of $K = 9$ tracks. The complete arrangement score is included in Appendix E.

In Figure 4, we observe some multi-track arrangement patterns that are common in practice. Purple blocks highlight a counterpoint relation between guitar track `A.G.1` and electric piano track `E.P.1`.

Green blocks show two guitar tracks with complementary orchestral functions: one melodic (A.G.1) and the other harmonic (A.G.2). Yellow blocks illustrate the metrical division between the string (Vlns.) and the brass (Br.) sections, with strings on the downbeat and brass on the offbeat. These patterns demonstrate a natural and cohesive multi-track arrangement by our system. Moreover, we see consistent accompaniment patterns echoing in both chorus parts that span over 30 bars (shown by red dotted boxes), while the latter adds a piano arpeggio track (Pno.) to enhance the musical flow. This demonstrates structured whole-song arrangement over extended music contexts.

# 5 Experiment

In this section, we evaluate the performance of our multi-track accompaniment arrangement system. Given that existing methods primarily focus on *lead sheet to multi-track* arrangement, we ensure a fair comparison by using the two-stage approach discussed in Section 4. In Section 5.1, we present the datasets used and the training details of our model. In Section 5.2, we describe the baseline models used for comparison. Our evaluation is divided into two parts: objective evaluation, detailed in Section 5.3, and subjective evaluation, covered in Section 5.4. For the single-stage *piano to multi-track* (Stage 2) and *lead sheet to piano* (Stage 1) arrangement tasks, we perform additional comparisons with various ablation architectures in Section 5.5 and 5.6, respectively.

## 5.1 Datasets and Training Details

We use two datasets to train the autoencoder and the style prior, respectively. The autoencoder is trained with Slahk2100 [25], which contains 2K curated multi-track pieces with 34 instrument classes in a balanced distribution. We discard the drum track and clip each piece into 2-bar segments with 1-bar hop size. We use the official training split and augment training samples by transposing to all 12 keys. The autoencoder comprises 19M learnable parameters and is trained with batch size 64 for 30 epochs on an RTX A5000 GPU with 24GB memory. We use Adam optimizer [19] with a learning rate from 1e-3 exponentially decayed to 1e-5. We use exponential moving average (EMA) [29] and random restart [7] to update the codebook with commitment ratio $\beta = 0.25$.

We use Lakh MIDI Dataset (LMD) [28] to train the prior model. It contains 170k music pieces and is a benchmark dataset for training music generative models. We collect 2/4 and 4/4 pieces (110k after processing) and randomly split LMD at song level into training (95%) and validation (5%) sets. We further clip each piece into 32-bar training excerpts (*i.e.*, $T = 16$ at maximum) with a 4-bar hop size. Our prior model has 30M parameters and is trained with batch size 16 for 10 epochs (600K iterations) on two RTX A5000 GPUs. We apply AdamW optimizer [22] with a learning rate of 1e-4, scheduled by a 1k-step linear warm-up followed by a single cycle of cosine decay to a final rate of 1e-6.

For model inference and testing, we consider two additional datasets: Nottingham [10] and WikiMT [44]. Both datasets contain lead sheets (in ABC notation) that are not seen during training or validation. Moreover, they cover diverse music genres including folk, pop, and jazz. When arranging a piece, we leverage control option 2 to set up the instrumentation. Without loss of generality, this control choice is randomly sampled from Slakh2100 validation/test sets. To arrange music longer than the prior model's context length (32 bars), we use windowed sampling [7], where we move ahead our context window by 4 bars and continue sampling based on the previous 28 bars. We apply nucleus sampling [13] with top probability p = 0.05 and temperature t = 6.

## 5.2 Baseline Models

We compare our system with three existing methods: PopMAG [30], Anticipatory Music Transformer (AMT) [33], and GETMusic [23]. PopMAG and GETMusic generate multi-track accompaniments from an input lead sheet based on a Transformer and a diffusion model, respectively. AMT is Transformer-based and it continues the accompaniment part from an input melody with starting accompaniment prompt. We provide detailed configurations in the following.

**PopMAG** is an encoder-decoder architecture based on Transformer-XL [6]. It represents multi-track music by sequential note-level tokenization and is fully supervised. The encoder takes a lead sheet as input and the decoder generates multi-track accompaniment auto-regressively. Since the model is not open source, We reproduce it on LMD with lead sheets extracted by [24] (melody) and [17] (chord).

Table 2: Objective evaluation results for *lead sheet to multi-track* arrangement (Section 5.3). All entries are of the form $\mathrm{mean} \pm \mathrm{sem}^s$, where $s$ is a letter. Different letters within a column indicate significant differences ($p < 0.05/6$) based on Wilcoxon signed rank test with Bonferroni correction.

| Model | Chord Acc. ↑ | DOA ↑ | Structure ↑ | Latency ↓ |
|-------|-------------|-------|-------------|-----------|
| Ours | $\mathbf{0.567} \pm 0.014^a$ | $\mathbf{0.300} \pm 0.004^a$ | $\mathbf{1.520} \pm 0.030^a$ | $0.461 \pm 0.005^b$ |
| AMT [33] | $0.446 \pm 0.013^{bc}$ | $0.294 \pm 0.006^a$ | $1.094 \pm 0.009^c$ | $6.320 \pm 0.212^d$ |
| GETMusic [23] | $0.423 \pm 0.012^c$ | $0.225 \pm 0.007^c$ | $1.243 \pm 0.017^b$ | $\mathbf{0.450} \pm 0.002^a$ |
| PopMAG [30] | $0.470 \pm 0.013^b$ | $0.270 \pm 0.007^b$ | $1.086 \pm 0.008^c$ | $0.638 \pm 0.013^c$ |
| Ground-Truth | - | $\mathbf{0.333} \pm 0.009$ | $\mathbf{1.980} \pm 0.019$ | - |

**Anticipatory Music Transformer (AMT)** is a decoder-only Transformer architecture with note-level tokenization. It introduces an "anticipation" method, where conditional tokens (melody and starting prompt) and generative tokens (accompaniment continuation) are interleaved to train a conditional generative model. Since our testing dataset does not provide ground-truth accompaniment, the starting prompt is given by the generation result (first 2 bars) of our system. We use the official implementation of the AMT model,[2] which is also trained on LMD.

**GETMusic** represents multi-track music as an image-like matrix resembling score arrangement, based on which a denoising diffusion probabilistic model is trained with a mask reconstruction objective. Given a lead sheet, it supports generating 5 accompaniment tracks using piano, guitar, string, bass, and drum or their subsets. In our experiment, we generate all 5 accompaniment tracks. We use the official implementation of the GETMusic model,[3] which is trained on internal data.

## 5.3 Objective Evaluation on Multi-Track Arrangement

We introduce four metrics to evaluate multi-track arrangement performance: *chord accuracy* [23, 30], *degree of arrangement* (*DOA*), *structure awareness* [42], and *inference latency* [30]. Among them, *chord accuracy* measures the multi-track harmony that reflects the fitness of the accompaniment to the lead sheet; *DOA* measures inter-track tonal diversity that reflects the creativity of the instrumentation. Both metrics demonstrate music cohesion at local scales. On the other hand, *structure awareness* measures phrase-level content similarity that reflects long-term structural coherence of the whole song. Finally, we use *inference latency* (in second/bar) to evaluate computational efficiency of each method. The detailed computation of each metric is provided in Appendix B. In Table 2, we compute ground-truth *DOA* using 1000 random pieces from LMD. We compute ground-truth *structure awareness* using 857 pieces in 4/4 from POP909 Dataset [38].

We randomly sample 50 pieces in 4/4 time signature from Nottingham and WikiMT respectively (100 pieces in total) to conduct experiment. The length of each piece ranges from 16 to 32 bars. We run our method and baseline models at each piece in 3 independent rounds, deriving 300 sets of multi-track arrangement samples. In Table 2, we report the evaluation results with mean value, standard error of mean (sem), and statistical significance computed by Wilcoxon signed rank test [43]. We find significant differences between our method and all baselines (p-value $p < 0.05/6$, using Bonferroni correction). In particular, our method outperforms in *chord accuracy*, *structure awareness*, and *DOA*, indicating the capability of arranging harmonious, structured, and creative accompaniments. The diffusion baseline outperforms in *inference latency* as it applies only 100 diffusion steps. Our method's efficiency is on par with it, while being 10 times faster than vanilla note-level auto-regression.

## 5.4 Subjective Evaluation on Multi-Track Arrangement

We also conduct a double-blind online survey to test music quality. Our survey consists of 5 evaluation sets, each containing an input lead sheet followed by 4 arrangement samples by our method and each baseline. Each sample is 24-32 bars long and is synthesized to audio at 90 BPM (~1 minute per sample). Both the set order and the sample order in each set are randomized. We request participants to listen to 2 sets and evaluate the musical quality based on a 5-point Likert scale from 1 to 5.

---

[2]`https://github.com/jthickstun/anticipation`
[3]`https://github.com/microsoft/muzic/tree/main/getmusic`

Table 3: Objective evaluation results for *piano to multi-track* arrangement (Section 5.5). All entries are of the form $\text{mean} \pm \text{sem}^s$, where $s$ is a letter. Different letters within a column indicate significant differences ($p < 0.05/6$) based on Wilcoxon signed rank test with Bonferroni correction.

| Prior | Faithfulness (stats.) ↑ | Faithfulness (latent) ↑ | DOA ↑ | NLL ↓ |
|---|---|---|---|---|
| Ours | $\mathbf{0.945} \pm 0.001^a$ | $\mathbf{0.215} \pm 0.005^a$ | $\mathbf{0.308} \pm 0.005^a$ | $\mathbf{0.411} \pm 0.004$ |
| Parallel | $0.937 \pm 0.002^b$ | $0.153 \pm 0.003^b$ | $0.233 \pm 0.005^c$ | $0.960 \pm 0.010$ |
| Delay | $0.915 \pm 0.004^c$ | $0.133 \pm 0.003^c$ | $0.207 \pm 0.005^d$ | $1.024 \pm 0.006$ |
| Random | $0.913 \pm 0.003^c$ | $0.113 \pm 0.003^d$ | $0.262 \pm 0.005^b$ | - |

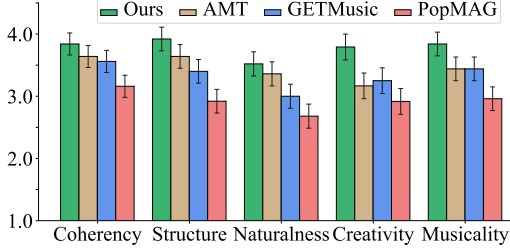
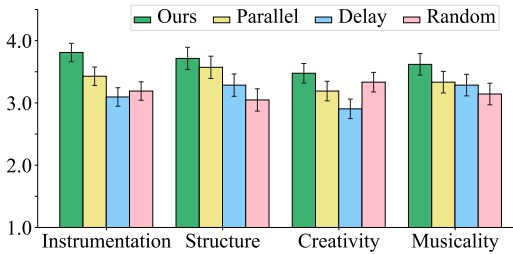

Figure 5: Subjective evaluation results on *lead sheet to multi-track* arrangement (Section 5.4).

Figure 6: Subjective evaluation results on *piano to multi-track* arrangement (Section 5.5).

The evaluation is based on 5 criteria: 1) *Harmony and Texture Coherency*, 2) *Long-Term Structure*, 3) *Naturalness*, 4) *Creativity*, and 5) *Overall Musicality*.

A total of 23 participants (8 female and 15 male) with diverse musical backgrounds have completed our survey, with an average completion time of 22 minutes. The mean ratings and standard errors, computed by within-subject (repeated-measures) ANOVA [31], are presented in Figure 5. Significant differences are observed across all criteria (p-value $p < 0.05$). Notably, our method outperforms all baselines on each criterion, aligning with the results from the objective evaluation.

## 5.5 Ablation Study on Style Prior Architecture

We now validate our design with the prior model by exclusively evaluating on the *piano to multi-track* arrangement task. Our prior model is based on a unique *layer interleaving* design, which enables multi-stream time series modelling with explicit stream-wise attention. We compare it with two other multi-stream architectures: 1) *Parallel*: summing up parallel code embeddings for joint language modelling [18], and 2) *Delay*: leveraging a 1-step delay code interleaving to catch implicit stream-wise dependency [4]. Both *Parallel* and *Delay* are trained under the same setup as our model. We additionally introduce 3) *Random*: a naive prior based on random template retrieval. The templates are sampled every 2 bars with shared instrumentation from the validation/test sets of Slakh2100.

We introduce two metrics to evaluate *piano to multi-track* arrangement: *faithfulness* and *degree of arrangement* (*DOA*). *Faithfulness* measures if the generated arrangement faithfully reflects the original content from the piano. It computes the similarity between i) the input piano, and ii) the *piano reduction* of the generated multi-track arrangement. In our case, we compute cosine similarity over two features: a *statistical* (*stats.*) pitch class histogram [45] and a *latent* texture representation [39], which emphasize tonal and rhythmic similarity, respectively. *DOA* measures the creativity as defined in Section 5.3. We also report the *NLL* loss for our model, *Parallel*, and *Delay*.

We conduct experiments using the test set of POP909 [39], which consists of 88 piano arrangement pieces. In our experiment, we use the first section of each piece, which contains 2 to 4 complete phrases totally spanning 24 to 32 bars. We use control option 3 to prompt our model, *Parallel*, and *Delay* with the same 2-bar *orchestral function* template (sampled from Slakh2100) and see how it is developed. We report mean value, standard error of mean (sem), and statistical significance in Table 3 and find significant differences in both *faithfulness* and *DOA*. We also conduct a subjective evaluation in the same setup as Section 5.4, with the results presented in Figure 6. Here we consider an additional criterion, *Instrumentation*, which reflects the well-formedness of the multi-

Table 4: Ablation study on alternative *lead sheet to piano* arrangement (*i.e.*, Stage 1) modules. Here we investigate the impact of Stage 1 on the entire two-stage system. Evaluation results are based on the final multi-track arrangement using respective Stage 1 modules.

| Two-Stage System | Chord Acc. ↑ | Structure ↑ | DOA ↑ |
|---|---|---|---|
| Ours (Stage 1 + Stage 2) | $\mathbf{0.567} \pm 0.014$ | $\mathbf{1.520} \pm 0.030$ | $\mathbf{0.300} \pm 0.004$ |
| Whole-Song-Gen [42] + Stage 2 | $0.509 \pm 0.015$ | $1.121 \pm 0.006$ | $0.277 \pm 0.006$ |

Table 5: Evaluation results for *lead sheet to piano* arrangement exclusively on Stage 1.

| Piano Arrangement Module | Chord Acc. ↑ | Structure ↑ |
|---|---|---|
| Ours (*Piano Texture Prior*) | $\mathbf{0.540} \pm 0.016$ | $\mathbf{1.983} \pm 0.147$ |
| Whole-Song-Gen [42] | $0.430 \pm 0.020$ | $1.153 \pm 0.180$ |

track arrangement. Significant differences are observed across all criteria (p-value $p < 0.05$). Overall, *Parallel* and *Delay* both fall short in performance because they assume a preset stream combination, while in our setting, both track numbers and choices of instruments are flexible. By explicitly modelling stream-wise attention, our *layer interleaving* design fits well to that generalized scenario.

### 5.6 Ablation Study on Piano Arrangement

Now we validate our choice for the *lead sheet to piano* arrangement module on the first stage of our two-stage system. Our choice is a *piano texture prior* as covered in Section 4. We conduct an ablation study by replacing it with the *Whole-Song-Gen* model [42], which, to our knowledge, is the only existing alternative that can handle a whole-song structure. The ablation study is conducted in the same setup as Section 5.3. In Table 4, we report *chord accuracy*, *structure awareness*, and *DOA* regarding the final multi-track arrangement results. We further compare our *piano texture prior* with *Whole-Song-Gen* exclusively on the piano accompaniment arrangement task. In Table 5, we report *chord accuracy* and *structure awareness* regarding piano arrangement for both models. Significant differences (p-value $p < 0.05$) are found in all metrics based on Wilcoxon signed rank test.

By comparing Table 4 and Table 5, we can see that a higher-quality piano arrangement generally encourages a more musical and creative final multi-track arrangement result. Specifically, the piano arrangement on Stage 1 lays the groundwork for (at least) chord progression and phrase structure for Stage 2, both of which are important for capturing the long-term structure in whole-song multi-track arrangement. Moreover, we see that our *piano texture prior* outperforms existing alternatives and guarantees a decent piano quality, thus being the best choice for our system.

## 6 Conclusion

To sum up, we contribute a music automation system for multi-track accompaniment arrangement. The main novelty lies in our proposed *style prior modelling*, a generic methodology for structured sequence generation with fine-grained control. By modelling the prior of disentangled style factors given content, we build a cascaded arrangement process: from lead sheet to *piano texture* style, and then from piano to *orchestral function* style. Our system first generates a piano accompaniment from a lead sheet, establishing the rough whole-song structure. It then orchestrates the piano accompaniment into a complete multi-track arrangement with band instrumentation. Extensive experiments show that our system generates structured, creative, and natural multi-track arrangements with state-of-the-art quality. At a higher level, we elaborate our methodology as *interpretable modular representation learning*, which leverages finely disentangled and manipulable music representations to tackle complex tasks with a compositional hierarchy. We hope our research brings new perspectives to broader domains of music creation, sequence data modelling, and representation learning.

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

# A    Implementation Details

## A.1    Autoencoder

The autoencoder consists of a VQ-VAE submodule and an overarching VAE. The encoder of the VQ-VAE consists of a 1-D convolutional layer of kernel size 4, stride 4, and 16 output channels, followed by a vector quantization block with codebook size 128. The decoder takes the concatenated latent codes and leverages two fully-connected layers (shape $128 \times 256$ and $256 \times 32$) for reconstruction. In the overarching VAE, Piano Encoder and Track Decoder are adapted from PianoTree VAE [40]. The encoder first applies a pitch-wise bi-directional GRU to summarize concurrent notes at time step $n$ and then applies a time-wise GRU to encode the full representation. The decoder mirrors the encoder structure with time- and pitch-wise uni-directional GRUs to reconstruct individual tracks. We use hidden size 256 in a single layer for pitch GRUs and 512 for time GRUs. The Track Separator is a 2-layer Transformer encoder with 8 attention heads, 0.1 dropout ratio, and GELU activation [12]. The hidden dimensions of self-attention $d_{\mathrm{model}}$ and feed-forward layers $d_{\mathrm{ff}}$ are 512 and 1024, respectively.

The autoencoder is trained with joint reconstruction loss for *orchestral function* (MSE) and individual tracks (cross entropy). The VQ-VAE is additionally regularized with latent loss and commitment loss with commitment ratio $\beta = 0.25$. The VAE is regularized with KL loss over all continuous factors ($\mathbf{c}_t$ and $\mathbf{z}_t^{1:K}$) based on KL annealing [41] with a ratio exponentially increasing from 0 to 0.5.

## A.2    Prior Model

The prior model consists of a 12-layer Context Encoder and a 12-layer Auto-Regressive Decoder. The latter is interleaved with another 12 track-wise Track Encoder layers. For each layer, we apply 8 attention heads, 0.1 dropout ratio, and GELU activation. We apply layer normalization before self-attention (*i.e.*, norm first). The hidden dimensions of self-attention $d_{\mathrm{model}}$ and feed-forward layers $d_{\mathrm{ff}}$ are 256 and 1024, respectively. We apply relative positional embedding [15] to Track Encoder so that two tracks initialized with identical instruments can still generate different content.

Our prior model is trained on the latent codes $\mathbf{c}_{1:T}$ and $s_{1:8T}^{1:K}$ inferred by a well-trained autoencoder on LMD. For discrete code $s$, we take the codebook indices and learn a new embedding.

# B    Objective Evaluation Metrics

## B.1    Degree of Arrangement

In multi-track arrangement, parallel tracks typically play a unique role to each other in the overall arrangement. We are interested in capturing the diversity and creativity inherent in these roles.

To achieve this, we consider the pitch class histogram [45] as a probability distribution $P$. Let $P_{t,k}$ be the distribution of the $t$-th bar in track $k$, and $P_t^{\mathrm{pn}}$ be that of the $t$-th bar in the *piano reduction*. Recall that in this paper we approximate the *piano reduction* of a multi-track piece by downmixing all tracks. Both $P_{t,k}$ and $P_t^{\mathrm{pn}}$ are 12-D vectors, describing tonality of individual tracks and the overall arrangement, respectively. We compute the KL divergence of each track to the *piano reduction*:

$$d_k = \frac{1}{T} \sum_{t=1}^{T} \mathbb{KL}(P_{t,k} \parallel P_t^{\mathrm{pn}}), \tag{5}$$

where $T$ is the total number of bars.

Interpreting $d_k$ in terms of KL divergence, we see it as the "excess surprise" from the overall arrangement (*piano reduction*) when track $k$ is played in isolation. A large $d_k$ indicates that track $k$ possesses a unique quality, such as a bass track playing the root or a counter-melody track focusing on tensions. Conversely, a small $d_k$ suggests that track $k$ serves as a foundational element in the arrangement, such as string padding that establishes the harmonic foundation.

If all $d_k$ values are small, it implies homogeneity across tracks and thus a low degree of arrangement. Conversely, if all $d_k$ values are high, it suggests a composition dominated by counterpoints, a scenario less common in pop music. A well-orchestrated piece typically exhibits a diverse range of $d_k$ values, encompassing both foundational and unique decorative tracks. We thus define the degree of

arrangement DOA as the standard deviation of $d_k$ for $k = 1, 2, \cdots, K$ across all tracks:

$$\text{DOA} = \sqrt{\frac{\sum_{k=1}^{K}(d_k - \overline{d})^2}{K}}, \tag{6}$$

where $\overline{d}$ is the mean. $K$ is the total number of tracks. To establish a reference point, we calculate the ground-truth DOA = 0.333 based on 1000 random pieces from the LMD dataset with at least 5 tracks. Within this context, a higher DOA signifies a more creative arrangement.

## B.2 Strcture Awareness

We introduce Inter-phrase Latent Similarity (ILS) from [42] to measure the structural awareness of long-term arrangement. ILS calculates the content similarity among same-type phrases (*e.g.*, chorus) versus the whole song. It leverages pre-trained disentangled VAEs that encode music notes into latent representations and then compare cosine similarities in the latent space. In our case, we compute ILS over the *piano reduction* of a generated arrangement since it contains the overall content. We apply the texture VAE [39] and obtain a latent texture representation $\mathbf{c}_t^{\text{txt}}$ for every 2-bar segment. For odd-numbered phrases, we repeat its final bar and pad it to the end of the phrase. Suppose there are $M$ different types of phrases in one piece and let $I_m$ be the set of segment indices in the type-$m$ phrase, ILS is defined as the ratio between same-type phrase similarity and global average similarity:

$$\text{ILS} = \frac{(\sum_{m=1}^{M}\sum_{i \neq j \in I_m} \cos(\mathbf{c}_i^{\text{txt}}, \mathbf{c}_j^{\text{txt}}))/(\sum_{m=1}^{M}|I_m|^2 - |I_m|)}{\sum_{1 \leq i \neq j \leq T} \cos(\mathbf{c}_i^{\text{txt}}, \mathbf{c}_j^{\text{txt}})/(T^2 - T)}, \tag{7}$$

where $|\cdot|$ is the cardinality of a set. $T$ is the number of 2-bar segments. When applying ILS, we use [5] to automatically label the phrase structure of a piece. To establish a reference point, we calculate the ground-truth ILS = 1.980 based on the POP909 dataset (with phrase annotation by human). Within this context, a higher ILS signifies saliency with long-term phrase-level structure.

## B.3 Chord Accuracy

We introduce chord accuracy from [30] to measure if the chords of the generated arrangement match the conditional chord sequence in the lead sheet. It reflects the harmonicity of the generated music and is defined as follows:

$$\text{CA} = \frac{1}{N_{\text{chord}}} \sum_{i=1}^{N_{\text{chord}}} \mathbf{1}_{\{C_i = \hat{C}_i\}}, \tag{8}$$

where $N_{\text{chord}}$ is the number of chords in a piece; $C_i$ is the $i$-th chord in the (ground-truth) lead sheet; and $\hat{C}_i$ is the aligned chord in the generated arrangement.

The original formulation in [30] considers chord accuracy for individual tracks. Given our system's capability to accommodate a variable combination of tracks, we opt for a broader evaluation for the overall arrangement. In our case, we extract the chord sequence of a generated arrangement with [17] and compare it in root and quality with ground-truth at 1-beat granularity, which is more rigorous.

## B.4 Orchestration Faithfulness

We measure the faithfulness of orchestration by the similarity between i) the *input piano* and ii) the *piano reduction* of the generated multi-track arrangement. Let $\mathbf{e}_t^{\text{in}}$ and $\mathbf{e}_t^{\text{pn}}$ be vector features derived from the $t$-th segment of the input and the reduction, respectively. Orchestration faithfulness OF is defined as follows:

$$\text{OF} = \frac{1}{T} \sum_{t=1}^{T} \cos(\mathbf{e}_t^{\text{in}}, \mathbf{e}_t^{\text{pn}}), \tag{9}$$

where $\cos(\cdot, \cdot)$ is cosine similarity. $T$ is the number of segments.

In our work, we select two options for vector feature $\mathbf{e}$. One is a statistical pitch class histogram [45], which is a 12-D vector describing pitch class distribution. The other is a latent 256-D texture representation learned by a pre-trained VAE [39]. Both features are general descriptors of the musical content with respective focus on tonal harmony and rhythmic grooves.

## C Experiment on Noise Weight $\gamma$

Continuing from Section 3.3, we compare different $\gamma$ values and see their impact on the model performance. When applying our model to *piano to multi-track* arrangement, $\gamma$ balances the force of a noisy factor added to the piano, which encourages a partial unconditional generation capability. The experimental settings are the same as Section 5.5. We evaluate the results based on *faithfulness* and *DOA*. In Table 6, we report mean value, standard error of mean (sem), and statistical significance computed by Wilcoxon signed rank test. By varying the $\gamma$ value, we observe a controllable balance between faithfulness and creativity. Specifically, a larger $\gamma$ encourages creativity (higher *DOA*) at the cost of *faithfulness*. If not mentioned otherwise, we use $\gamma = 0.25$ for experiments in this paper.

Table 6: Objective evaluation results on the impact of noise weight $\gamma$ in Appendix C.

| Noise Weight $\gamma$ | Faithfulness (stats.) ↑ | Faithfulness (latent) ↑ | DOA ↑ |
|---|---|---|---|
| $\gamma = 0$ | $\mathbf{0.946} \pm 0.001^a$ | $\mathbf{0.228} \pm 0.005^a$ | $0.300 \pm 0.005^c$ |
| $\gamma = 0.25$ | $0.945 \pm 0.001^{ab}$ | $0.215 \pm 0.005^b$ | $0.308 \pm 0.005^{bc}$ |
| $\gamma = 0.5$ | $0.944 \pm 0.001^b$ | $0.187 \pm 0.004^c$ | $0.320 \pm 0.006^{ab}$ |
| $\gamma = 1$ | $0.936 \pm 0.002^c$ | $0.127 \pm 0.003^d$ | $\mathbf{0.325} \pm 0.007^a$ |

## D Online Survey Specifics

We distribute our survey via SurveyMonkey.[4] Our survey consists of 5 sample sets for both the *lead sheet to multi-track* and the *piano to multi-track* arrangement tasks (10 sets in total). Each sample is 24-32 bars long and is synthesized to audio at 90 BPM using BandLab[5] with the default soundfont. Each participant listens to 2 sets (in random order) and the mean time spent is 22 minutes. Figure 7 shows the sample pages of our survey with instructions to the participants.

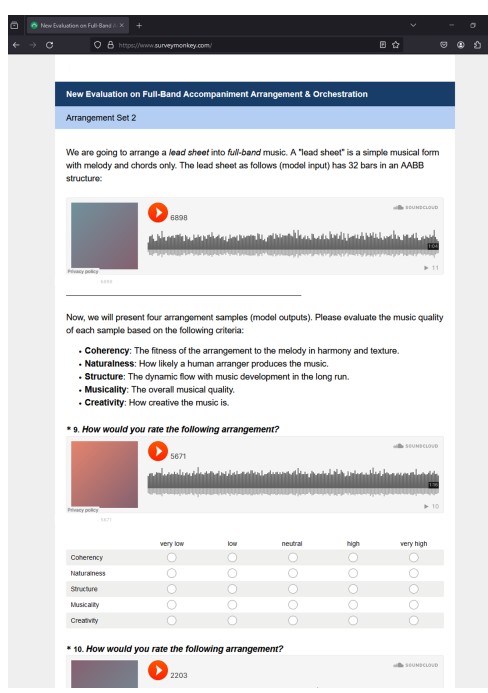

(a) *Lead sheet to multi-track* arrangement.

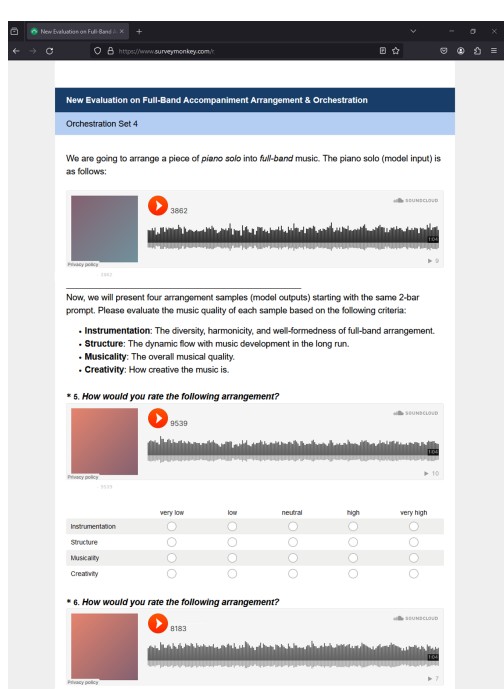

(b) *Piano to multi-track* arrangement.

Figure 7: Screenshots of survey pages and instructions of our online survey.

---

[4] https://www.surveymonkey.com
[5] https://www.bandlab.com/

# E Example on Structured Arrangement

We demonstrate an example of accompaniment arrangement by our proposed system. The input lead sheet is a complete pop song shown in Section E.1. Our system first arranges a piano accompaniment for the whole song, which is shown in Section E.2. The piano score is then orchestrated into a multi-track arrangement with customized instrumentation, which is shown in Section E.3.

## E.1 Lead Sheet

We use our system to arrange for *Can You Feel the Love Tonight*, a pop song by Elton John. As shown in Figure 8, the entire song is 60 bars long and it presents a structure of i4A8B8B8x4A8B8B8O4, where i, x, O, A, and B each refer to intro, interlude, outro, verse, and chorus.

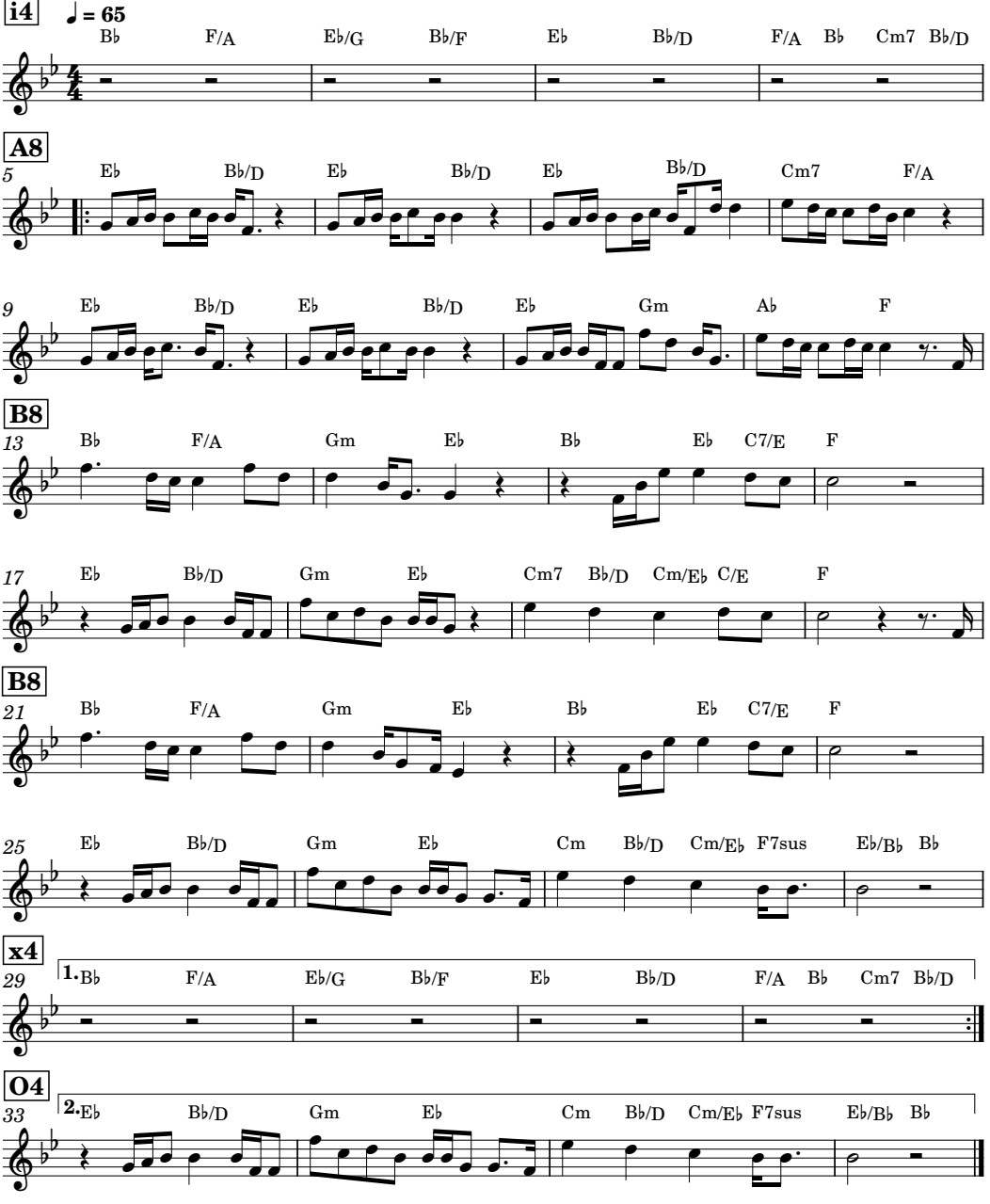

Figure 8: Lead sheet for pop song *Can You Feel the Love Tonight*.

## E.2 Piano Arrangement

The piano arrangement result is shown from Figure 9 to Figure 10. It roughly establishes a whole-song structure and lays the groundwork for band orchestration at the next stage. Demo audio for the piano score is available at `https://zhaojw1998.github.io/structured-arrangement/`.

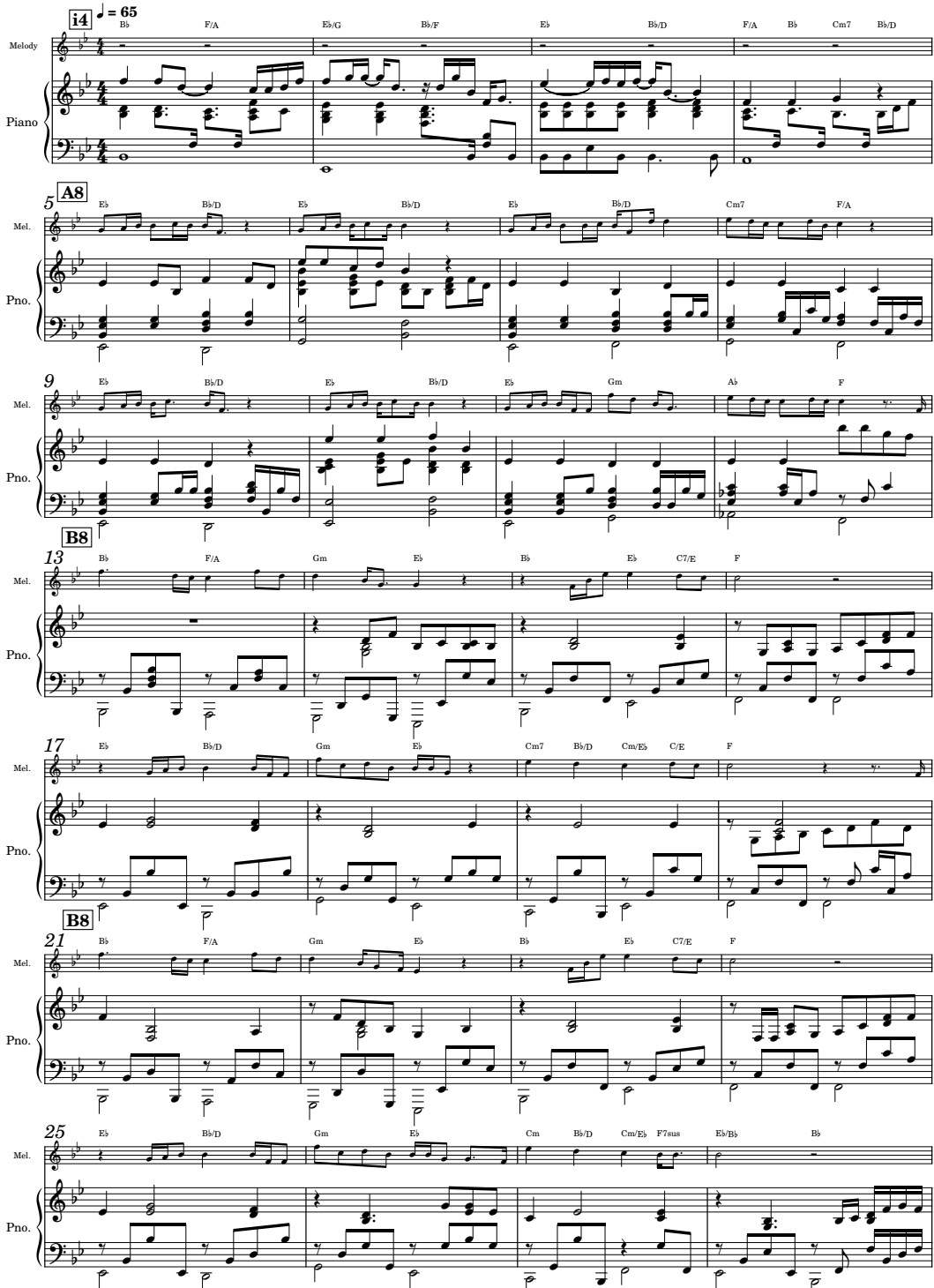

Figure 9: Piano arrangement score (page 1).

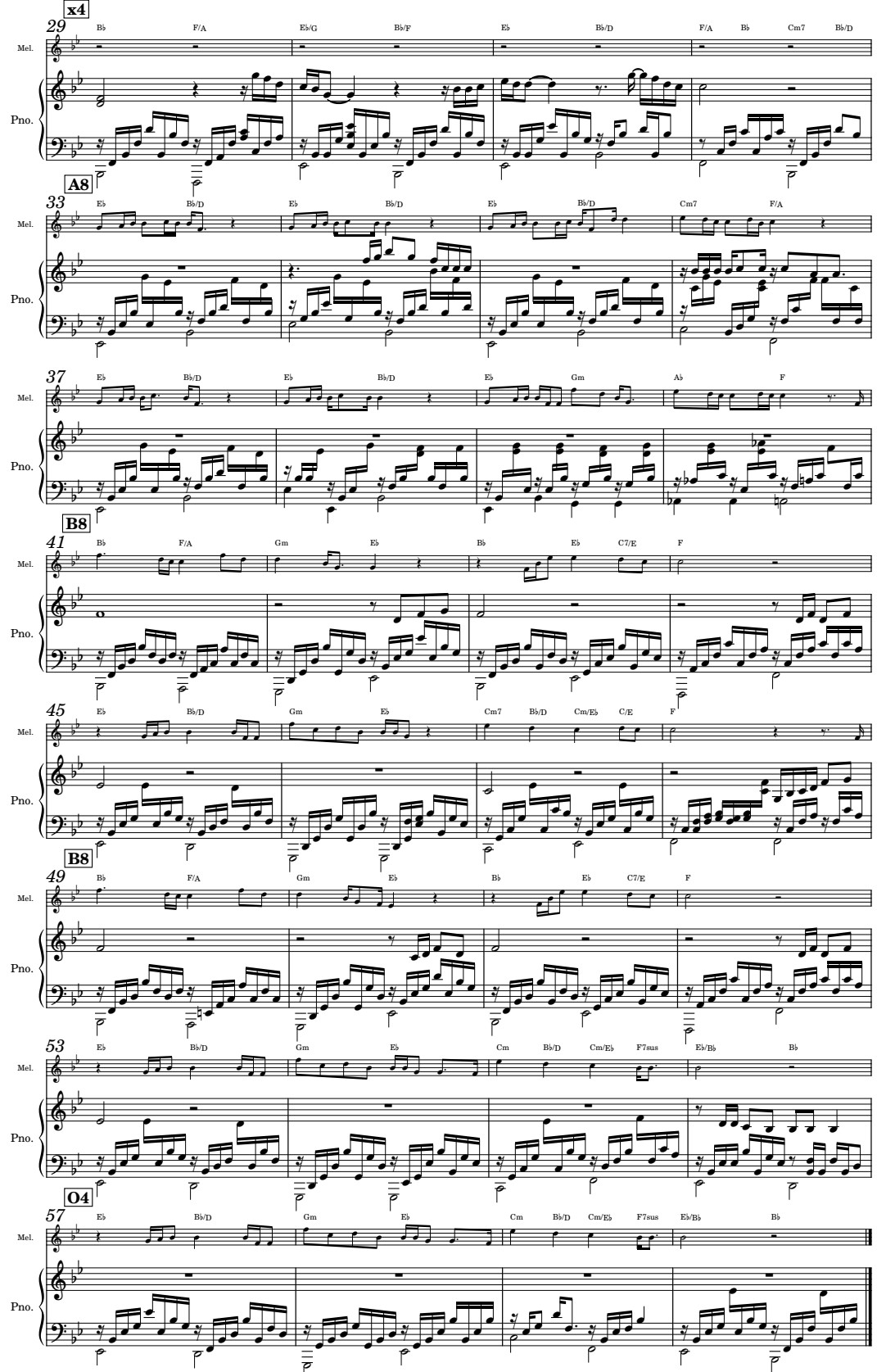

Figure 10: Piano arrangement score (page 2, last page).

### E.3 Multi-Track Arrangement

The multi-track arrangement is shown from Figure 11 to Figure 15. We customize the instrumentation as celesta, acoustic guitars (2), electric pianos (2), acoustic piano, violin, brass, and electric bass in a total of $K = 9$ tracks. We can see that the structure of the accompaniment follows the lead sheet. Demo audio is available at `https://zhaojw1998.github.io/structured-arrangement/`. More detailed analysis on this arrangement demo is covered in Section 4.

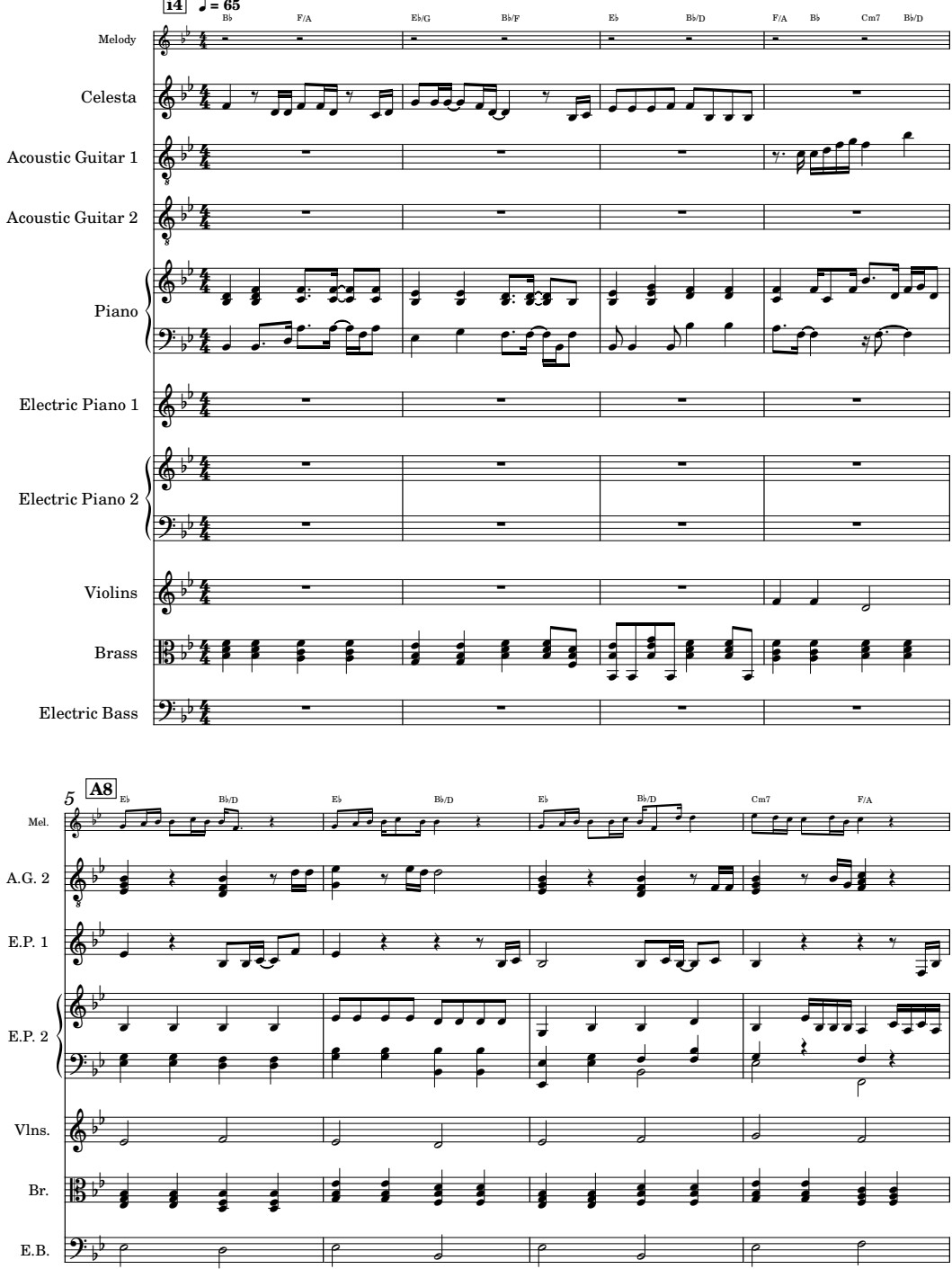

Figure 11: Multi-track arrangement score (page 1).

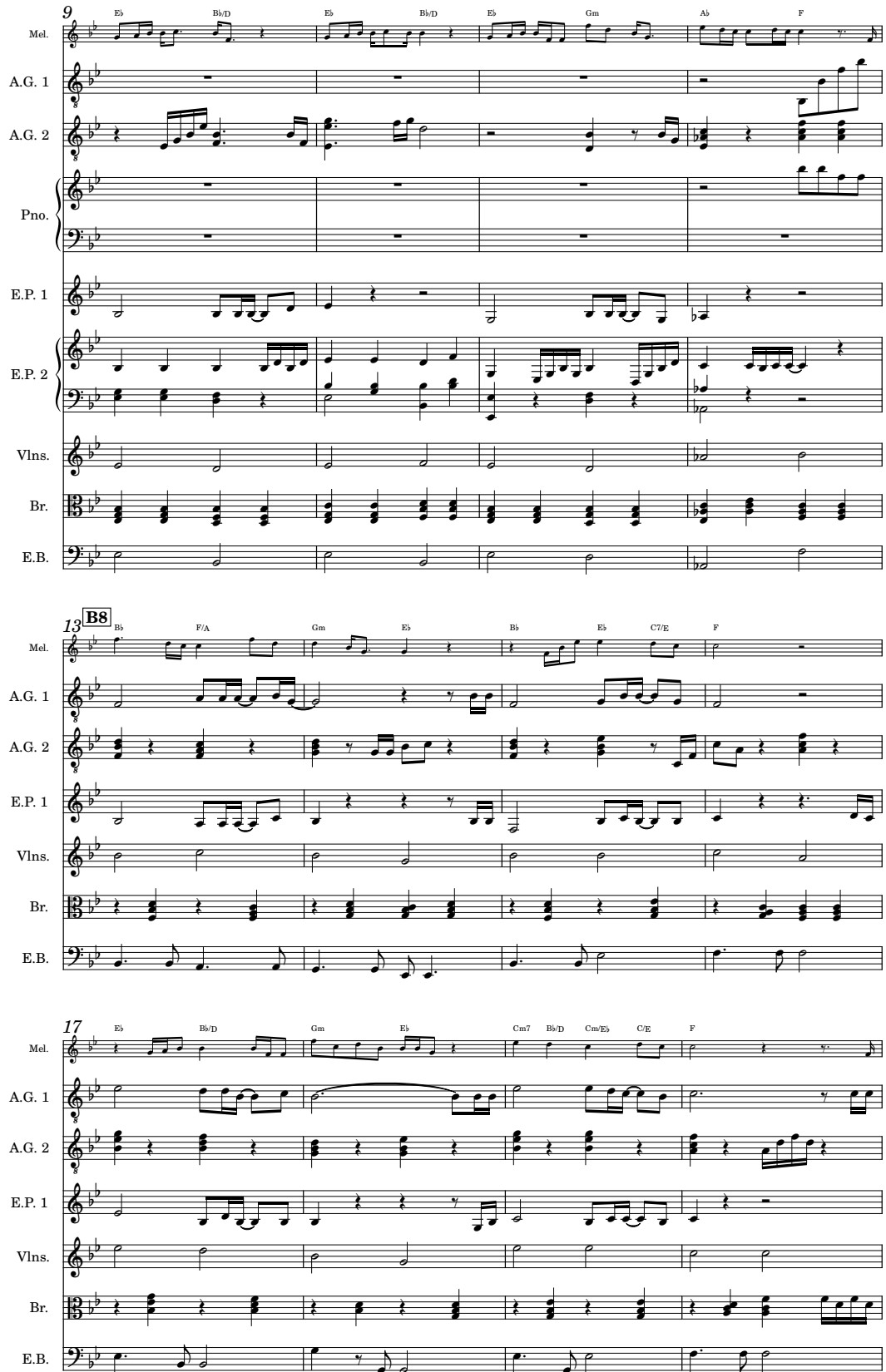

Figure 12: Multi-track arrangement score (page 2).

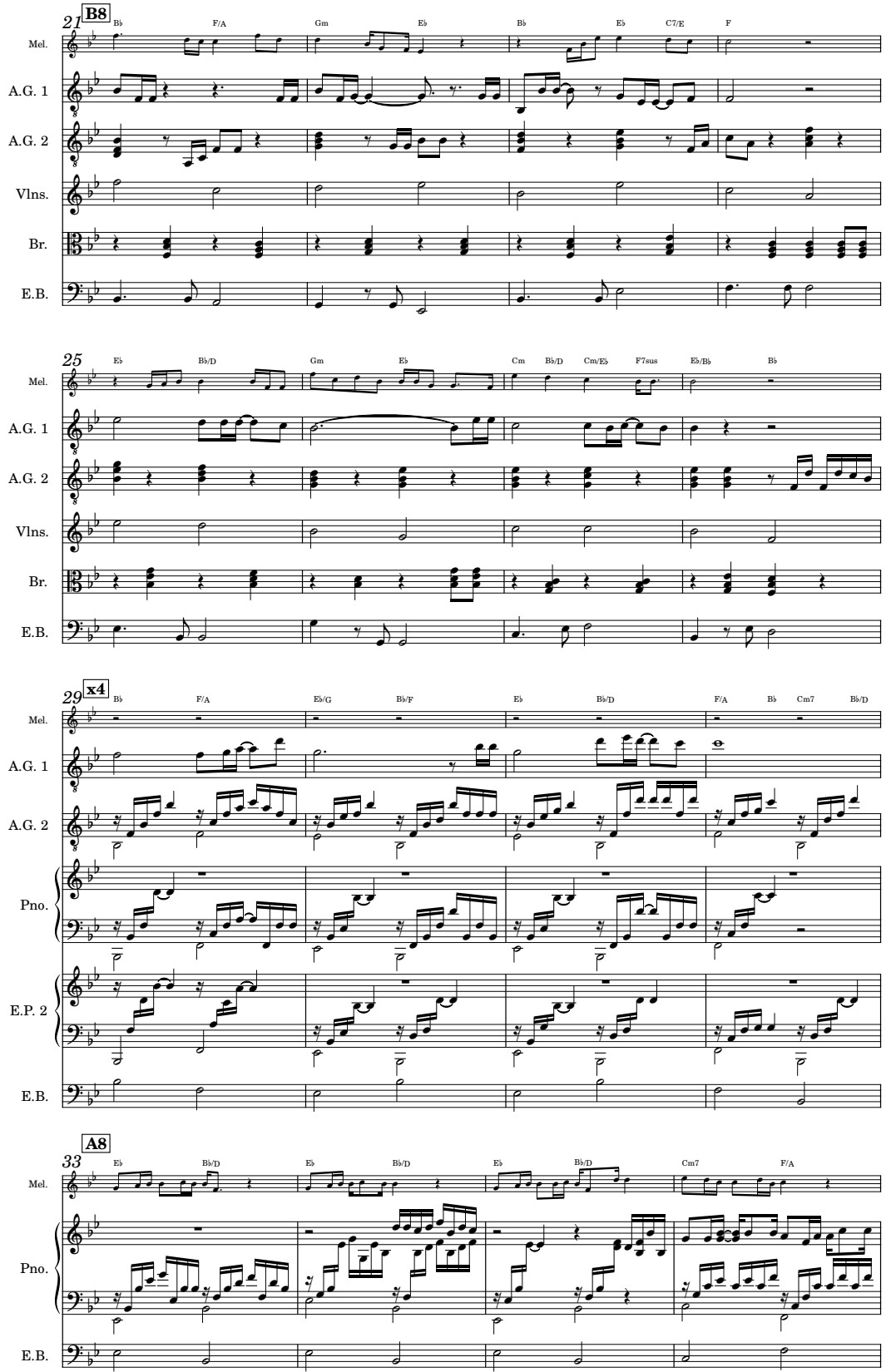

Figure 13: Multi-track arrangement score (page 3).

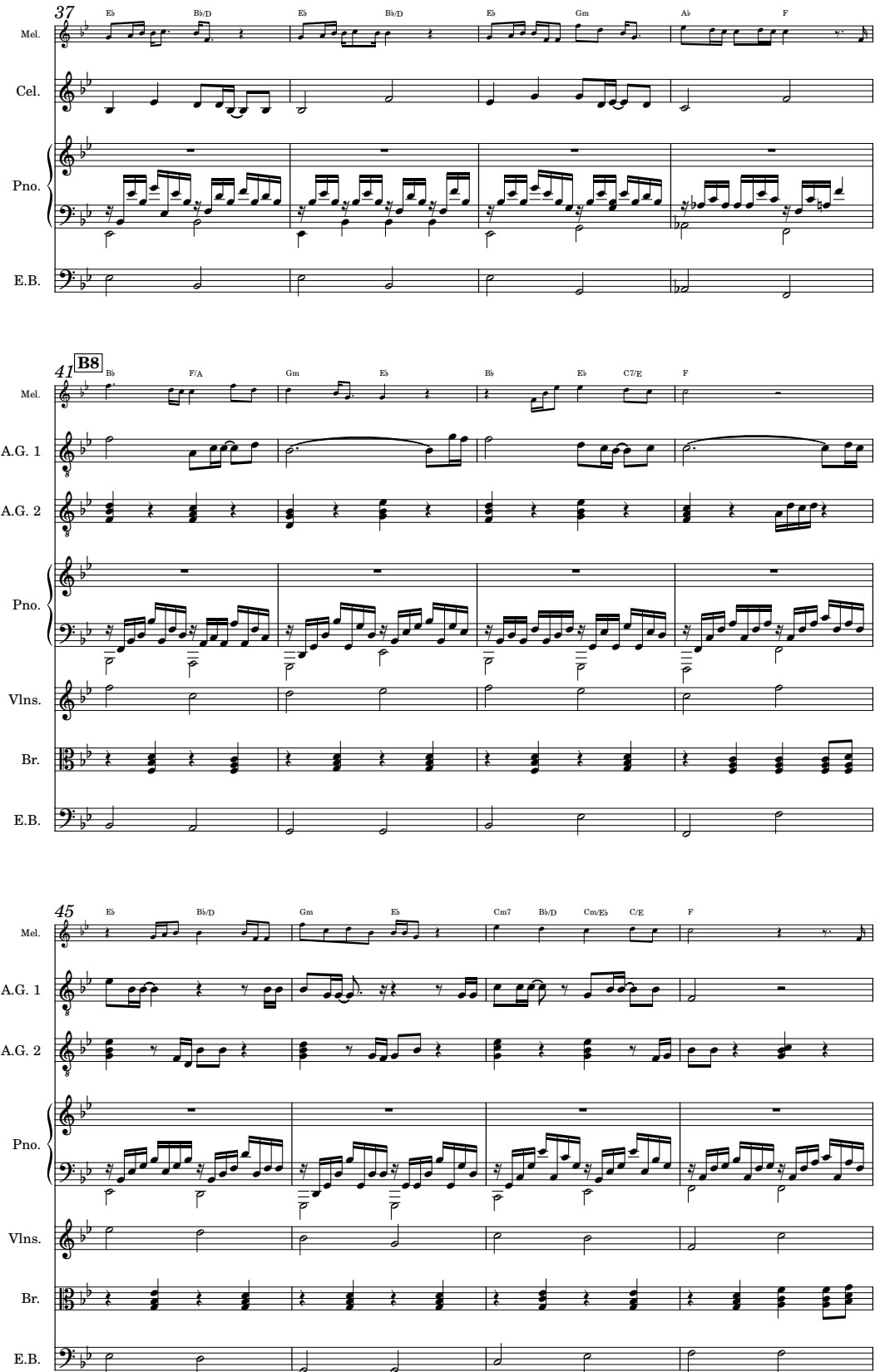

Figure 14: Multi-track arrangement score (page 4).

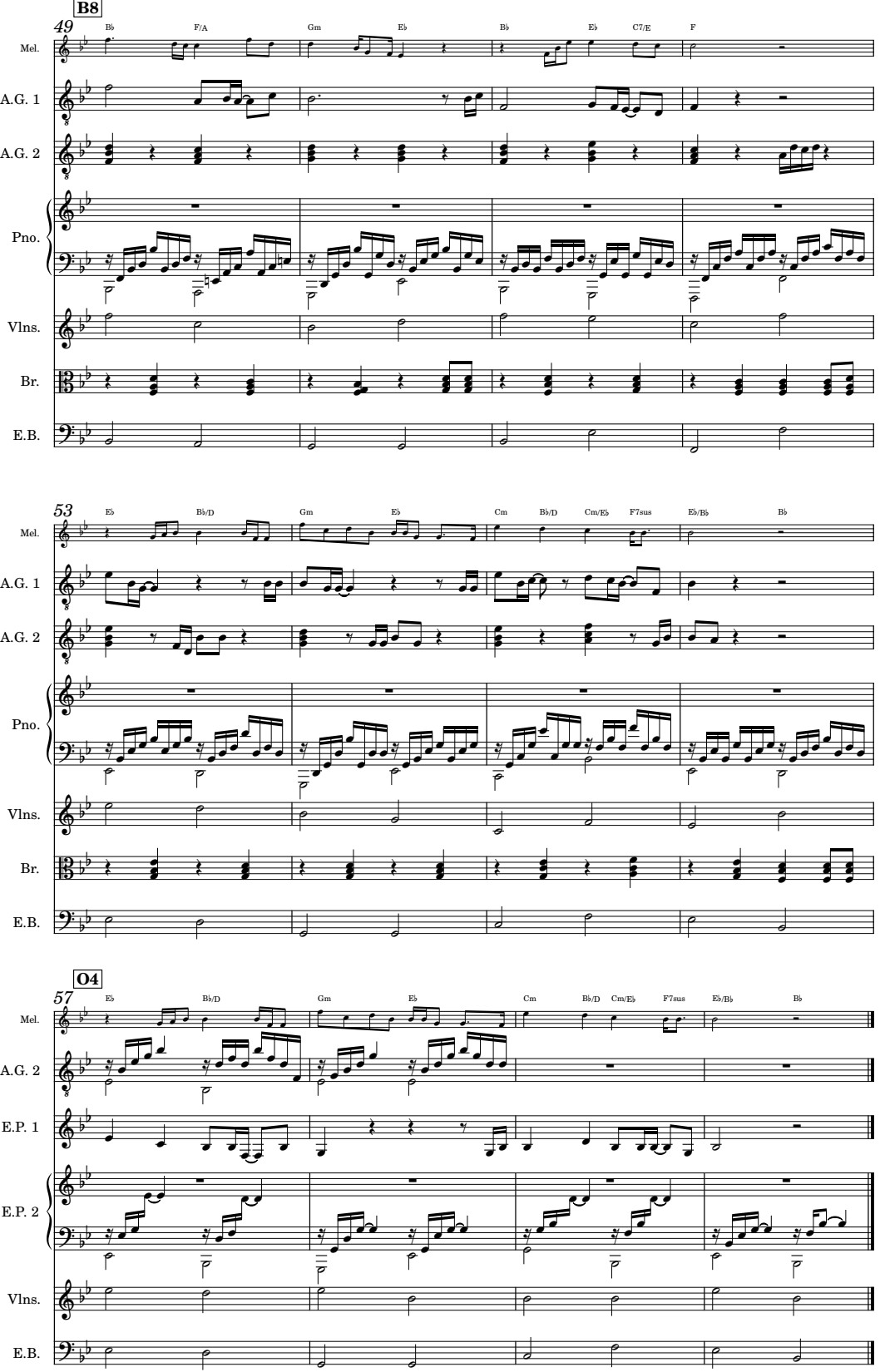

Figure 15: Multi-track arrangement score (page 5, last page).

# F Limitation

We propose a two-stage system for whole-song, multi-track accompaniment arrangement. In the context of this paper, we acknowledge that our current system exclusively supports tonal tracks in quadruple meters while disregarding triple meters, triplet notes, and drums. However, we perceive this as a technical limitation rather than a scientific challenge. We also acknowledge that our current system primarily emphasizes the composition level, thereby omitting the modelling of MIDI velocity, dynamic timing, and MIDI control messages. Consequently, the generated results do not encompass performance MIDI and may lack expressive qualities. Nevertheless, we believe that our composition-centric work serves as a solid and vital foundation for further advancements in those specific areas, thus facilitating the development of enhanced techniques and features. As a pioneering work, our system is the foremost accomplishment in solving whole-song multi-track accompaniment arrangement, characterized by flexible controllability on track number and choice of instruments.

# G Broader Impacts

Our multi-track accompaniment arrangement system, which incorporates *style* to generate accompaniment, is designed to enhance originality and creativity. It serves as a platform for human-AI co-creation, where *the user provides content-based material* (in our case, lead sheet) that remains fundamentally original, while *the AI agent infuses style, enriches the form, and enhances creativity*. Our system therefore empowers musicians to explore new musical ideas and expand their creative boundaries. This approach also allows for rapid mock-up with different styles and arrangements, fostering an environment where innovation and artistic expression can thrive.

However, we acknowledge the need to address potential risks. The accessibility of our system may inadvertently lead to excessive reliance on automation, potentially impeding the development of fundamental skills among musicians. Additionally, widespread adoption of the system may contribute to the homogenization of music, threatening the distinctiveness and individuality that are crucial to artistic expression. We recognize that our datasets predominantly feature contemporary Western music, which introduces a cultural bias that could limit the diversity of generated compositions.

