# OpenReview forum: "Structured Multi-Track Accompaniment Arrangement via Style Prior Modelling"
_NeurIPS.cc/2024/Conference — NeurIPS 2024 poster_

### Official Review · Reviewer_AQ7F · 2024-07-06

**Soundness:** 3
**Presentation:** 3
**Contribution:** 3
**Rating:** 7
**Confidence:** 4

**Summary:**

This paper introduces a novel music AI system that creates multi-track accompaniments from a lead sheet by leveraging disentangled style factors to enhance context coherence, creativity, and computational efficiency. The proposed two-stage process begins with generating a piano arrangement using piano texture styles, followed by orchestrating multiple tracks by incorporating orchestral function styles. The system employs vector quantization and a multi-stream Transformer for modeling orchestration styles, significantly improving arrangement quality and providing flexible control over various music genres and compositional levels.

**Strengths:**

1. The paper addresses the significant task of Multi-Track Accompaniment Arrangement, making notable improvements in the field.

2. The authors have diligently constructed a comprehensive framework in both the experimental design and methodology, contributing valuable insights and data to the field.

**Weaknesses:**

1. The introduction lacks clarity, with key terms and relationships not sufficiently defined before introducing mathematical notations and formulations. This obscures the motivation and makes it difficult to understand, especially as the abstract outlines challenges related to coherency, creativity, and efficiency that are not clearly addressed in the introduction.

2. The contributions of the paper, as discussed between lines 46-60 and 85-94, appear to be incremental rather than significant, primarily building upon work from AccoMontage.

3. There is a consistent omission of motivation when new concepts are introduced. For instance, the motivation behind formulating the multi-track accompaniment arrangement task is not made clear in the introduction after discussing style representations. Similarly, the reasons behind specific mechanisms in sections 3 and 4 are not elaborated upon.

**Questions:**

1. Why does concatenating style factors potentially contradict the structure of existing conditions as mentioned on line 25?

2. The paper discusses conflicts when combining style factors in sequence generation tasks; could these be resolved using different methods like diffusion models?

3. How do the initial discussions on introducing style into long-term generation relate to the paper’s main theme of multi-track accompaniment arrangement?

4. What prompted the formulation of the problem between lines 32-36 in paragraph 3, and how is time t determined?

5. Is there a distinction between 'piano reduction' and 'piano track'? The terminology varies between italics and normal text; does this signify a difference?

6. What is the rationale behind using VQ-VAE and VAE to encode orchestral functions and piano reductions in section 3.2?

7. What motivations underlie the construction of the Multi-Track Orchestral Function Prior in section 3.3?

8. Can you explain what inputs the proposed model receives and provide comparisons to highlight its unique features, as seen in Figure 4?

9. How does the model handle both ABC notation and Midi, and is there a conversion of ABC notation into the representation discussed in section 3.1?

10.Why is the computational efficiency of this model less optimal compared to [20]?

11. How are the 'style' factors controlled and tested in the experiments?

**Limitations:**

Limitations are addressed.

---

> ### Author Rebuttal · Authors · 2024-08-06
>
> Thank you for your review and constructive feedback! We acknowledge that some terms in our paper may not be sufficiently clear (**W1**). We also recognize our shortcomings in providing adequate motivation to justify our design choices (**W3**). Please allow us to first respond to the relevant points raised in **W2** and **Questions**. We hope this could address your concerns. We will then comprehensively incorporate these clarified ideas into our manuscript to provide sufficiently clear introduction and well-justified motivation.
>
> **W2. Significance of Contribution**
>
> Our proposed *style prior modelling* is not built upon AccoMontage. It is an original methodology that effectively addresses long-term, structured conditional sequence generation. AccoMontage addresses piano arrangement only, while our method is more scalable to solve more challenging practical problems. Building upon *style prior modelling*, we introduce the first whole-song, multi-track accompaniment arrangement system, which supports variable music length ($T$) and controllable instrumental tracks ($K$).
>
> **Q1. Why  concatenating style factors may contradict the structure of existing conditions**
>
> A: The general problem that we study is *conditional sequence generation*. The input is a conditional sequence $\mathbf{c}\_{1:T}$ and the output is an observational sequence $\mathbf{o}\_{1:T}$. *We assume there underlies a style factor sequence $\mathbf{s}\_{1:T}$*, which can render/arrange the content of $\mathbf{c}\_{1:T}$ to realize $\mathbf{o}\_{1:T}$. By *style prior modelling* of $\mathbf{s}\_{1:T}$, we aim to improve interpretability, controllability, and performance. We note that the existing condition $\mathbf{c}\_{1:T}$ often implies a long-term structure. The style factor $\mathbf{s}\_{1:T}$, if not structurally aligned with $\mathbf{c}\_{1:T}$, will lead to incoherent results.
>
> **Q2. Can diffusion models address this?**
>
> A: Diffusion models and Transformers can be distinct implementations of the same methodology. The context dependency of diffusion models differs from that of Transformers and we will explore this in our future research.
>
> **Q3. Relation to accompaniment arrangement**
>
> A: Accompaniment arrangement is a typical task of *long-term* conditional sequence generation. The input is a lead sheet that implies a verse-chorus structure of the whole song. The output is the accompaniment, and the style factor regards the sequential *form* of the accompaniment. In this paper, we consider multi-track, whole-song arrangement, where it is challenging to maintain both track cohesion and structural coherence.
>
> **Q4. Prompt of the formulation in lines 32-36**
>
> A: Continuing from Q1 & Q3, our general idea is to model $\mathbf{s}$ conditional on $\mathbf{c}$. We thus come to the formulation of $p(\mathbf{s}\_{1:T}^{1:K} | \mathbf{c}\_{1:T})$. Superscript $k=1, 2, \cdots K$ denotes track indices of multi-track arrangement. Subscript $t = 1, 2, \cdots, T$ denotes component segments of the whole song. We consider each segment as a 2-bar music snippet, which is a proper scale for learning content/style factor representations.
>
> **Q5. *Piano reduction* vs 'piano track'**
>
> A: We use *piano reduction* to denote the overall content from a multi-track piece. We will unify all instances of *piano reduction* into italic text. On the other hand, a ‘piano track’ simply refers to a general track played by piano.
>
> **Q6. Rationale for VQ-VAE and VAE**
>
> A: We use VAE to learn the nuanced content from the *piano reduction* because existing works have shown VAE’s effectiveness for music representation learning. We further choose VQ-VAE for the *orchestral function* because common patterns of orchestral tracks (like syncopation, arpeggio, etc.) can naturally be categorized as discrete variables. Moreover, VQ-VAE learns a discrete latent space that facilitates a prior model to mount on.
>
> **Q7. Motivations for Multi-Track Orchestral Function Prior**
>
> A: By constructing Multi-Track Orchestral Function Prior, we *recover the underlying style factor sequence* that can further render/arrange the input content into multi-track accompaniment. This design is more interpretable and controllable while also adhering to music composition practice. Experiments also demonstrate superior performance against non-prior baselines.
>
> **Q8. Model inputs in Figure 4**
>
> A: In Figure 4, the model receives two inputs: a lead sheet shown by the “Mel” staff, and a set of instruments (user control) shown by the rest staff labels. The output is the accompaniment in the rest staves. The length of lead sheet determines $T$ and the number of instruments determines $K$.
>
> **Q9. ABC notation**
>
> A: ABC is essentially a score notation for lead sheet, which can be converted into MIDI and the representation in Section 3.1.
>
> **Q10. Computational Efficiency of GETMusic**
>
> A: GETMusic is a diffusion model with only 100 diffusion steps, thus achieving notable computational efficiency.
>
> **Q11. How the style factors are controlled and tested**
>
> A: The control over style factors is covered in lines 173-176. Firstly, user can customize the instruments to steer the Multi-Track Orchestral Function Prior. Moreover, a starting prompt can optionally be provided. In our experiment, these control choices are randomly sampled from Slakh2100 test set. The effectiveness of $\mathbf{s}\_{1:T}$ is manifested from the quality of the resulting $\mathbf{o}_{1:T}$. Experiments show that *our method achieves the top chord, structure, and DOA scores for long-term arrangement*, which demonstrates the effectiveness of style factors and the superiority of *style prior modelling*.

---

> > ### Comment · Reviewer_AQ7F · 2024-08-13
> >
> > Thank you for your detailed and thoughtful response to my review. Your clarifications, particularly regarding the significance of your contribution and the motivations for the methodologies employed, provide a deeper understanding of your work. I look forward to seeing these enhanced explanations incorporated into the revised manuscript, as they will undoubtedly strengthen its clarity and justification.

---

### Official Review · Reviewer_ANaR · 2024-07-10

**Soundness:** 4
**Presentation:** 2
**Contribution:** 3
**Rating:** 6
**Confidence:** 4

**Summary:**

The paper presents a style transfer-based music accompaniment generation system. It starts with the lead melody and fleshes out the accompaniment tracks based on various high-level information, such as instrument type and rhythmic structure. The main goal of the proposed model is to be able to generate coherent structure in long music sequences, while they are also cohesive. Model architecture adopts VQVAE baselines to learn latent representations while the sequence learning part is done via transformer models applied to tracks and time dimensions orthogonally. The objective and subjective test scores look promising.

**Strengths:**

- The paper presents reasonable intermediate representations of symbolic music, piano reduction and orchestral function, which are used to learn discrete embeddings by the VQVAE model.

- The proposed model is evaluated by both objective and subjective manners, where it outperformed the other methods.

**Weaknesses:**

- The description of the proposed model is not organized well, so it is difficult to understand.

- Some high-level choices the authors made are not well justified, such as discrete embeddings vs. continuous, Gaussian noise vs. other noise, etc.

**Questions:**

- In Fig. 1, the autoencoder takes the piano reduction information as its first input. I imagine it would correspond to the lead sheet. Does it mean that during the training time the output of the autoencoder is the full sheet with all tracks? Then this isn't an autoencoder, right? The precise training procedures are not detailed enough in this manuscript.

- If the input PN is the full music during training, how does the model generalize to the test-time scenario when the input has to be only the lead sheet?

- The manuscript seems to be based on the concept of 1 segment = 8 beats. Is this correct? Any rationale behind this choice?

- Gaussian noise is used to regularize the model. But if it's to deal with the domain shift between piano reduction representations, wouldn't there be a different type of noise that's more suitable, i.e. more discrete random variables?

- It appears that the "s" embeddings contain important information about the track to be generated. Since it's learned in an unsupervised way, it's a little hard to imagine how these embeddings are more interpretable than other existing conditioning vectors as the authors claim.

**Limitations:**

- Basically it's not clear how Fig. 1 and Fig. 2 are combined to form the pipeline described in Fig 3. Colored figures, shades, and hashes show the authors' effort in describing this method, but it's not entirely clear to me.

- It appears that the system takes the user input to designate which instrumental track to generate based on the instrument embedding. It's a nice feature, but it also means that, for evaluation, somebody has to come up with a nice orchestration (i.e., which instruments go well given the lead melody). Was this critical selection of instruments done by the authors to generate the test sequences? Unless I missed this part, it has to be clearly mentioned, as it can affect the performance of the model (in comparison to other methods that don't have such a concept).

---

> ### Author Rebuttal · Authors · 2024-08-06
>
> Thank you for your review and constructive feedback! Please allow us to first respond to the points raised in **Limitations** and **Questions**. We hope this could address your concerns. We will then comprehensively incorporate these clarified ideas into our manuscript to provide sufficiently clear organization (**W1**) and well-justified motivation (**W2**).
>
> **L1. Elaboration on the model architecture**
>
> A: In this paper, we introduce a two-stage system to address the challenging task of multi-track accompaniment arrangement. Stage 1 arranges lead sheet into piano, and Stage 2 arranges piano into multi-track full sheet. Fig. 3 in the manuscript demonstrates the two-stage pipeline, which sequentially arranges piano/full sheets using respective style factors. The technical method of this paper mainly focuses on Stage 2, where we use the autoencoder (Fig. 1) to disentangle *piano reduction* $\mathrm{pn}[\mathbf{x}]$ (content factor) and *orchestral function* $\mathrm{fn}[\mathbf{x}]$ (style factor) from full sheet $\mathbf{x}$. In Stage 2, $\mathrm{pn}$ is given as condition and we propose the prior model (Fig. 2) to infer $\mathrm{fn}$.
>
> **L2. User control on instrument designation**
>
> A: Our system does rely on user designation of instrumental tracks, and we see it as an intuitive and handy control. In practice, users can try out preset instrument ensembles (e.g., pop band with piano, guitars, and bass) without added burden. In our experiment, without loss of generality, this control choice is randomly sampled from Slakh2100 test set (mentioned in line 222).
>
> **Q1. Input/output of autoencoder and training detail**
>
> A: The autoencoder takes two inputs: the *piano reduction* $\mathrm{pn}[\mathbf{x}]$  and the *orchestral function* $\mathrm{fn}[\mathbf{x}]$. The output is the full sheet $\mathbf{x}$. Note that both $\mathrm{pn}[\mathbf{x}]$ and $\mathrm{fn}[\mathbf{x}]$ are deterministic transforms from $\mathbf{x}$. This is an inductive bias for content/style disentanglement. The ultimate input is $\mathbf{x}$ and the training is based on a self-supervised reconstruction objective. We can see similar autoencoder designs in other disentanglement works [1] as well.
>
> **Q2. How the model generalizes to the test-time scenario**
>
> A: At test time, the autoencoder takes the piano arrangement from Stage 1 as its first input.
>
> **Q3. The rationale behind segment scale**
>
> A: Yes. We consider 1 segment = 8 beats. This is a proper scale (i.e., neither too short nor too long) to capture composition structures. Existing studies on music representation learning have also applied the 8-beat scale in their work [2,3].
>
> **Q4. The rationale for Gaussian noise against discrete ones**
>
> A: We note that the *piano reduction* encoder learns *continuous* content representations instead of discrete ones (vector quantization is not applied here). Music content can be nuanced and thus better described with continuous variables. We hence see Gaussian noise as a natural choice for the domain shift regularization.
>
> **Q5. How the $\mathbf{s}$ embeddings are interpretable**
>
> A: The $\mathbf{s}$ embeddings are encoded from *orchestral function* $\mathrm{fn}[\mathbf{x}]$, which essentially describes the *form*, or *layout*, of multi-track music $\mathbf{x}$. It contains rhythmic intensity information, telling the model where to put more notes and where to keep silent. When learning $\mathbf{s}$ embedding from $\mathrm{fn}[\mathbf{x}]$, we apply vector quantization because common rhythmic patterns (like syncopation, arpeggio, etc.) can naturally be categorized as discrete variables. Moreover, VQ-VAE learns a discrete latent space that facilitates a prior model to mount on.
>
> [1] Z. Wang, et al. Audio-to-symbolic arrangement via cross-modal music representation learning. ICASSP 2022.
>
> [2] A. Robert, et al. A hierarchical latent vector model for learning long-term structure in music. ICML 2018
>
> [3] R. Yang, et al. Deep music analogy via latent representation disentanglement. ISMIR 2019.

---

### Official Review · Reviewer_Y3LG · 2024-07-13

**Soundness:** 4
**Presentation:** 4
**Contribution:** 3
**Rating:** 7
**Confidence:** 4

**Summary:**

The paper suggests creating multi-instrument accompaniment by using the piano reduction and the instrument note density (referred to as the 'Orchestration Function') as bootstrap representations. By effectively applying VAE and an autoregressive sequence generation framework, the paper demonstrates the high potential and effectiveness of the proposed approach for learning musically structured accompaniment.
The paper also proposed a 'layer interleaving architecture' that processes the orchestration codec by alternating between the time axis and the track axis.
The paper compares with previous works with objective and subjective evaluation and shows its validity.

**Strengths:**

The strength of this paper, I believe, lies in the factorization of accompaniment generation by leveraging the fact that piano reduction and 'orchestration function' can be freely produced as middle-level features from multi-track MIDI. By separating each process, as the authors claim, it becomes applicable to scenarios where users can exert control. Additionally, by dividing the modeling of each accompaniment's prior and the modeling of detailed notes, the efficiency of learning for each model is expected to have increased.

**Weaknesses:**

The proposed methodology utilizes piano reduction as a condition, so the quality of the final orchestra accompaniment is expected to vary depending on the quality of the piano reduction. Upon listening to the provided examples, it is evident that notes not present in the piano reduction were added, confirming that the proposed model's role extends beyond merely rearranging the notes of the piano reduction. However, there needs to be a discussion on the impact of using piano reduction on the model's high evaluation compared to other models. Additionally, while it was mentioned in L168 that another module was used, a more detailed explanation of how the piano reduction was generated is necessary.

**Questions:**

Q1. From my understanding, the 'orchestration function' in Eq. 1 eliminates pitch information. Thus, I expect the priors (${s_t}$) to provide note-density-like bare information. What do you expect to be encoded in $s$? Also, as a follow-up question, what kinds of correlation are expected along with $s_t$, and why does it make sense to train a sequential prediction model for $s$? I'm not sure about the expected role of the prior model.

**Limitations:**

I generally agree with the authors about the limitation presented in Appendix E.

---

> ### Author Rebuttal · Authors · 2024-08-06
>
> Thank you for your review and thoughtful feedback! We hope the following will address your concerns:
>
> **Weakness: Discussion on the impact of the piano reduction quality**
>
> A: We introduce *piano reduction* as an intermediate representation from the input lead sheet to the final orchestra arrangement. Intuitively, *piano reduction* is a hierarchical and more abstract planning of the final orchestra. The orchestrator at Stage 2 may add notes, but it still falls within the scope implied by the *piano reduction*.
>
> To formally investigate the impact of *piano reduction*, we conduct an ablation study by replacing the original piano arranger with the *Whole-Song-Gen* model [1], which, to our knowledge, is the only existing alternative that can handle a whole-song structure. The ablation study is conducted in the same setting as Section 5.3. We report objective evaluation results regarding the final orchestra’s *chord accuracy*, *structure awareness*, and *degree of arrangement* (*DOA*) as follows:
> | | *Chord Acc*  $\uparrow$ |*Structure*  $\uparrow$ |*DOA* $\uparrow$ |
> | -------- | ------- | ------- | ------- |
> | Ours			| $0.567 \pm 0.014^a$ | $1.520 \pm 0.030^a$ | $0.300 \pm 0.004^a$ |
> | Using *Whole-Song-Gen*| $0.509 \pm 0.015^b$ | $1.121 \pm 0.006^b$ | $0.277 \pm 0.006^b$ |
>
>
> We can observe that *Whole-Song-Gen* at Stage 1 generally deteriorates the quality of the final orchestra. To see why this happens, we further compare *Whole-Song-Gen* with our original piano arranger exclusively on the piano arrangement stage. We report objective evaluation results regarding the piano’s *chord accuracy* and *structure awareness* as follows:
> | | *Chord Acc*  $\uparrow$ |*Structure*  $\uparrow$|
> | -------- | ------- | ------- |
> | Original piano arranger| $0.540 ± 0.016^a$ | $1.983 ± 0.147^a$ |
> | *Whole-Song-Gen*| $0.430 ± 0.020^b$ | $1.153 ± 0.18^b$ |
>
> By comparing the two tables, we can see that a higher-quality *piano reduction* generally encourages a more musical and creative final orchestra result. Particularly, *piano reduction* *lays the groundwork for (at least) chord progression and phrase structure*, both of which are important for capturing the long-term structure in whole-song arrangement.
>
> Overall, we confirm that the *piano reduction* is an abstract planning of the final orchestra result. Thus its quality is positively correlated to the orchestra quality. Meanwhile, we see that *our current piano arranger significantly outperforms existing alternatives and guarantees decent piano quality*, thus being the best choice for our model. Additionally, both the piano arranger at Stage 1 and the orchestrator at Stage 2 can be applied as independent modules to address respective subtasks.
>
> [1] Z. Wang, et al. Whole-song hierarchical generation of symbolic music using cascaded diffusion models. ICLR 2024.
>
> **Q1. What information is encoded in $\mathbf{s}$**
>
> A: The $\mathbf{s}$ embeddings are encoded from *orchestral function* $\mathrm{fn}[\mathbf{x}]$, which essentially describes the *form*, or *layout*, of orchestra sheet $\mathbf{x}$. It contains rhythmic intensity information, telling the model where to put more notes and where to keep silent.
>
> **Q2. What kind of correlation to be expected along with $\mathbf{s}\_{1:T}$**
>
> A: $\mathbf{s}\_{1:T}$ encodes the sequential *form* of orchestra sheet. As $t$ goes from $1$ to $T$, *with the development of music*, we may see new rhythm patterns being introduced and new instrumental tracks being activated (e.g., in Figure 4 of the manuscript, the piano track is activated in the second half of the piece to mark increased atmosphere). More importantly, the lead sheet (and piano reduction) $\mathbf{c}\_{1:T}$ implies a verse-chorus structure of the whole song. Our prior model infers $\mathbf{s}\_{1:T}$ conditional on $\mathbf{c}\_{1:T}$, thus guaranteeing the structural alignment between $\mathbf{s}\_{1:T}$ and $\mathbf{c}\_{1:T}$.

---

### Official Review · Reviewer_iTts · 2024-07-16

**Soundness:** 4
**Presentation:** 3
**Contribution:** 4
**Rating:** 8
**Confidence:** 4

**Summary:**

The authors introduce a new model for generating multi-track accompaniment given a lead sheet. While being strictly a two-stage method (first generate a piano arrangement from the lead sheet and *then* generate the accompaniment), the authors' contribution is focused on the 2nd stage, and they rely on existing modules for the 1st stage. Their proposed method features two key components: **first**, a VQ-VAE submodule whose aim is to learn (quantized) representations of "orchestral functions" (orchestral functions being 1-D time-series representations of a track, with each element representing the sum of notes active in each segment, more on that in weaknesses), and, **second**, a transformer-based "mixing" VAE which combines the representations of the VQ-VAE (i.e., the representations of the orchestral functions) with learnt representations of "piano reductions" of each song (derived by averaging all individual tracks) to generate latent representations which are used to synthesize the full song. A key novelty of their model is the "mixing" part of the VAE, with the decoder utilizing interleaved time-wise and track-wise cross-attention layers -- meaning, that each output "sees" information both from past frames in its own track and from other tracks. Their system compares favorably to 3 standard baselines taken as-is from prior work or reproduced.

**Strengths:**

Overall, barring some presentation issues discussed below, this is a well-written paper that features state-of-the-art results and enough novelty to make it relevant for both the broader generative AI community (given the difficulty of generating long-form, complex music which the authors tackle with hierarchical modelling) and the more niche music-generation community. This is why I am leaning positive towards accepting the paper. I will mention some more of its strengths in detail: + The authors target a challenging task which is to generate a multi-track arrangement given a lead sheet. Even though they "only" focus on the 2nd stage of that pipeline, namely the generation of the multi-track arrangement given a piano arrangement, it is still a valid contribution. + The idea to disentangle the piano reduction and orchestral function and combine them in the way they do is interesting and novel. This also provides a nice element of controllability. + The fact that the model can be trained in "self-supervised" fashion (i.e. without manual annotations) is commendable. + The authors have done a thorough job of documenting experimental parameters (data, GPU runtime, etc.) and submit code to reproduce their experiments. + The authors have accounted for different sources of variability and provided multiple "knobs" to twist (and counteract domain mismatches) through the clever use of multiple "positional" embeddings (Fig. 2).

**Weaknesses:**

The main weaknesses of the paper are primarily in terms of presentation, rather than methodology or evaluation. My comments are mainly meant to help the authors improve their presentation.

* Clarity of music concepts: Sometimes, the authors introduce or reference concepts which are not straightforward without domain knowledge. Examples of that are: * The orchestral function of x is introduced as a column sum of activation indicators over the different columns of X. Given that the columns of X represent MIDI pitches, this would mean that at each frame, the authors are summing the notes that are active. Yet, in p.3, l.115, they write that this indicator function "counts each note onset position as 1". Why onset only? This would mean that each note is counted once. But the equation states $x_t^k>0$, which appears to be evaluated for each frame (thus, not only at the onset of each note, but for its entire duration). Further, they state that "it [orchestral function] describes the rhythm and groove of each track". It is not at all clear to me why this orchestral function should describe the rhythm? It only appears to describe the amount of notes active at a given time. It's even less clear why it should describe the groove, given the subjective nature of "groove". It is okay that the authors attempt to give a layman's explanation to their method in this part, but it is also recommended that they better describe those concepts (and do so in a way that would be acceptable even to those that are not familiar with concepts from music theory).

 * There are other terms, such as "counterpoint", "phrase", etc. which will not be evident in an audience not familiar with them. While it is understandable that in a music generation paper, there will be a lot of domain language, it would be useful if the authors commented on their importance in their discussion of results. For example, the ILS metric measures the similarity within a music phrase vs the rest of the song. What does it mean that the authors' method does better than the baselines in this metric? That it produces more coherent phrases that are more easily distinguishable from the rest of the song? Adding a sentence to clarify the importance of each result (in, as far as this is possible, layman's terms) would improve the readability and outreach of the paper

* Comparability to baselines: The authors compare to three established baselines. However, with the exception of PopMAG, which the authors reproduce on part of the data they used to train their own model, the other two baselines are not strictly comparable. AMT requires ground-truth accompaniment (arguably a limitation of that method compared to the authors'), and GETMusic was trained on a different data (and additionally handles different instrumental tracks). While this is not a no-go, and is common practice for generative methods in general, it should be mentioned in 5.2.

* The description of *interleaving* layers in the autoencoder -- according to the authors themselves, a major novelty of the work -- does not feature as prominently in their text as it should (given its stated importance), and could be improved in terms of clarity. Specifically, it is not entirely clear what the "Track Encoder" does in Fig. 2. Is it a standard residual "transformer" (it is not clear from A.2 if a residual connection is there) layer that essentially combines information across tokens? It would be beneficial to describe this in 3.3 using simple equations like in 3.2. One important open question is why this track encoder, if it is indeed based on self-attention, is different than the decoder? Specifically, why does one integrate information in the time-axis and the other on the track-axis? It would be important to show the dimensions here. I guess you have: $(B, K, N, P)$, where B is the batch size, K the number of tracks, N the duration of each note, and P the number of notes (I guess N&P are eventually downsampled from their original dimensions). So the inter-track encoder would operate on tensors $(B\cdot N, K, P)$ and then the intra-track decoder would operate on tensors $(B\cdot K, N, P)$? All this of course would be done in autoregressive fashion, so it would go from $\{1...N\}$. It would be nice to show this using the notation of the authors. * The VQ-VAE is described (A.1) and portrayed (Fig. 1) as having a convolutional encoder with a stride of 4 (thus a downsampling of 4) and a fully-connected decoder. It would be useful to show how this fully-connected decoder mitigates the downsampling of the encoder to have an output size equal to the input. Minor comments: + p.7, l.240: Best substitute "denoising process" with denoising diffusion probabilistic model (if this is what this meant), as the term denoising can be ambiguous

**Questions:**

There is only one question, regarding the interleaving of layers, but since it is also a weakness, I have included it there instead.

**Limitations:**

The authors have adequately discussed their limitations.

---

> ### Author Rebuttal · Authors · 2024-08-06
>
> Thank you for your review and constructive feedback! We value your thorough and insightful comments and will revise our manuscript based on these suggestions to improve the presentation. We also hope the following could address your existing concerns:
>
> **Weakness 1. Clarification of music concepts**
>
> A: Yes, we see the importance of clarifying music-related terms to make our paper more comprehensible. Certain terms (like 'groove') may have slightly varied definitions in music and AI research. We will use them in the research context and make the best effort to explain them well. We provide an explanation to the raised points as follows:
>
> (Count each note onset position as 1) - In this paper, we represent MIDI tracks using the *modified* pianoroll representation [1]. Instead of spreading out the full duration of a note, each non-zero $\mathbf{x}_t^k =1, 2, \cdots, 32$ denotes the duration of a note at timestep $t$ and track $k$. Hence the indicator function $\mathbf{x}_t^k>0$ recovers the note onset positions.
>
> (*Orchestral function*, rhythm, and groove) - Our intuition with *orchestral function* is inspired by the "grooving pattern" introduced in [2], the essence of which is to use note onset densities (in our case, intensities) to describe rhythms. Groove admittedly has a subjective nature, but it is generally associated with the rhythm. In our paper, we mention "groove" in the same context as [2] and we will clarify this in our future revision.
>
> [1] Z. Wang, et al. Learning interpretable representation for controllable polyphonic music generation. ISMIR 2020.
>
> [2] S.-L. Wu and Y.-H. Yang. The jazz transformer on the front line: Exploring the shortcomings of ai-composed music through quantitative measures. ISMIR 2020.
>
> **Weakness 2. Clarification with other terms**
>
> A: Yes, we see the benefits of interpreting the essence of specific terms that may otherwise be too abstract to comprehend. Your interpretation of the ILS metric is well aligned with our intention. We will follow this example to elaborate on the other terms.
>
> **Weakness 3. Comparability to baseline**
>
> A: We will clarify our model’s comparability to each baseline in Section 5.2.
>
> **Weakness 4. (Q1.) Interleaving Layers**
>
> We will improve the presentation of layer interleaving in our future revision. Regarding model dimensions, we have tensors $(B, K, T, D)$, each being batch size, track number, time frames (downsampled), and feature dimension. The inter-track encoder operates on tensor $(B \times T, K, D)$, and the autoregressive decoder on $(B \times K, T, D)$. We note that the inter-track encoder is not autoregressive, but a standard residual Transformer Encoder layer. This gives the user full control of initializing the number ($K$) and instruments of tracks.
>
> We will supplement the missing details in the VQ-VAE framework and revise the ambiguous terms. Thank you again for your constructive advice on the paper presentation!

---

### Decision · Program_Chairs · 2024-09-25

**Decision:**

Accept (poster)

**Comment:**

**Summary**: The paper introduces a new system for generating multi-track accompaniments from a lead sheet using a style prior modelling approach. The method involves a two-stage process: generating a piano arrangement and then orchestrating multiple tracks. The system leverages vector quantization and a multi-stream Transformer to model orchestration styles, enhancing generative capacity and providing flexible control.

**Strengths**: The paper presents a unique method for disentangling piano reduction and orchestral function, providing a novel element of controllability. The proposed model is technically robust and has a well-executed evaluation. The authors have documented experimental parameters thoroughly and provided code for reproducibility. The method addresses a challenging task in music AI, making significant contributions to the broader generative AI community and the niche music-generation community.

**Weaknesses**: The reviewers noted that the paper could benefit from clearer explanations of concepts and better organization of the model description (e.g., what are interleaving layers). Some baselines used for comparison are not strictly comparable, which should be clarified in the paper. Some high-level design choices, such as discrete embeddings and Gaussian noise, need better justification.

**Recommendation**: The reviewers unanimously agree that the paper should be accepted. As such, it is recommended for acceptance conditioned on minor revisions being implemented to address the presentation issues and provide clearer justifications for high-level design choices. The unique approach and state-of-the-art performance make it a valuable contribution to the field of music AI.